# Ice Crystal Characterization in Cirrus Clouds: A Sun-tracking Camera System and Automated Detection Algorithm for Halo Displays

Linda Forster[1], Meinhard Seefeldner[1], Matthias Wiegner[1], and Bernhard Mayer[1,2]

[1]Chair of Experimental Meteorology, Ludwig-Maximilians-Universität, München, Germany
[2]Institut für Physik der Atmosphäre, Deutsches Zentrum für Luft- und Raumfahrt, Oberpfaffenhofen, Germany

*Correspondence to:* Linda Forster (linda.forster@physik.lmu.de)

**Abstract.** Halo displays in the sky contain valuable information about ice crystal shape and orientation: e.g. the 22° halo is produced by randomly oriented hexagonal prisms while parhelia (sundogs) indicate oriented plates. HaloCam, a novel sun-tracking camera system for the automated observation of halo displays is presented. An initial visual evaluation of the frequency of halo displays for the ACCEPT (Analysis of the Composition of Clouds with Extended Polarization Techniques) field campaign from October to mid-November 2014 showed that sundogs were observed more often than 22° halos. Thus, the majority of halo displays was produced by oriented ice crystals. During the campaign about 27% of the cirrus clouds produced 22° halos, sundogs or upper tangent arcs. To evaluate the HaloCam observations collected from regular measurements in Munich between January 2014 and June 2016, an automated detection algorithm for 22° halos was developed, which can be extended to other halo types as well. This algorithm detected 22° halos in about 2% of the time for this dataset. The frequency of cirrus clouds during this time period was estimated by co-located ceilometer measurements using temperature thresholds of the cloud base. About 25% of the detected cirrus clouds occurred together with a 22° halo which implies that these clouds contained a certain fraction of smooth, hexagonal ice crystals. HaloCam observations complemented by radiative transfer simulations and measurements of aerosol and cirrus optical thickness provide a possibility to retrieve more detailed information about ice crystal roughness. This paper demonstrates the feasibility of a completely automated method to collect and evaluate a long-term database of halo observations and shows the potential to characterize ice crystal properties.

## 1 Introduction

Cirrus clouds represent about 30% of the global cloud coverage (Wylie et al., 1994) and play an important role in the Earth's energy budget. They consist of small non-spherical ice crystals, which scatter and absorb solar radiation and emit thermal infrared radiation. Depending on which of the two effects dominates, cirrus clouds have either a cooling or a warming effect on climate. The radiative properties of cirrus clouds are governed not only by their optical thickness and ice crystal effective radius, but also depend crucially on the ice crystal shape and orientation (Yi et al., 2013; Wendisch et al., 2007). Better knowledge of shape, surface roughness, and orientation of ice crystals in cirrus clouds would therefore help to improve estimates of the radiative forcing of cirrus clouds as well as satellite retrievals of cirrus optical properties as discussed by Yang et al. (2015) and

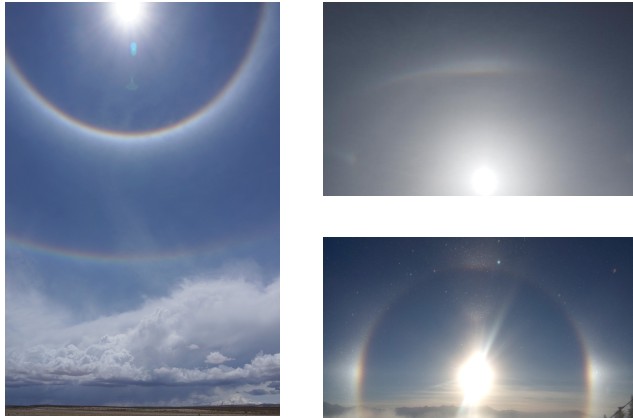

**Figure 1.** Left: A bright 22° halo or circumscribed halo with infralateral arc below, Salar de Uyuni, Bolivia, October 2, 2014 (photograph by Leonhard Scheck). Top right: upper tangent arc with faint sundogs in Munich, Germany, April 1, 2014. The halo displays are faint due to the high aerosol concentration in the air. Bottom right: a 22° halo with upper tangent arc and bright sundogs on Mt. Hohe Salve, Austria, January 18, 2016 (photograph by Volker Freudenthaler).

references therein.

Halo displays are produced by hexagonal ice crystals with smooth faces via refraction and reflection of sunlight. The formation of halo displays has already been described by Pernter and Exner (1910), Wegener (1925), Minnaert (1937) and by a number of later publications (Tricker, 1970; Greenler, 1980; Tape, 1994; Tape and Moilanen, 2006). One of the most common displays
is the 22° halo which appears as a bright ring around the sun at a scattering angle of about 22° and is formed by randomly oriented hexagonal ice crystals. Further frequently observed halo displays are the parhelia of the 22° halo, commonly called sundogs, which are caused by sunlight refracted by horizontally oriented hexagonal plates. Hexagonal ice crystal columns with their long axis oriented horizontally form another halo type: the upper and lower tangent arcs. Their shape changes with the solar elevation. When the sun is close to the zenith both the upper and lower tangent arc merge to the circumscribed halo.
Fig. 1 shows examples of the most frequent halo displays. The left image depicts a bright 22° or circumscribed halo with a rare infralateral arc below. A faint upper tangent arc and two faint sundogs are shown on the upper right image and very bright sundogs with a faint 22° halo and small upper tangent arc are displayed on the lower right image. Halos are not only beautiful optical displays but also contain valuable information about ice particle shape and orientation. Recent publications showed that the brightness contrast of the 22° halo in ice crystal scattering phase functions is related to the aspect ratio and surface
roughness of the crystals (van Diedenhoven, 2014). Quantitative analysis of e.g. frequency of occurrence or brightness contrast of halo displays, can therefore help to determine ice crystal properties, such as shape, surface roughness and orientation in cirrus clouds.

Probably the first reported photometric measurements of halo displays were performed by Lynch and Schwartz (1985) who took a photo of a 22° halo around the moon with a Kodak Plus-X pan film camera. After digitizing the photo, the halo brightness

and width was analyzed and compared with theoretical values to infer information about ice crystal size and shape.

In order to exploit the information content of halo displays, continuous long-term observations of cirrus clouds are required. In the 1990's many observations have been collected by amateur halo-observing networks (Pekkola, 1991; Verschure, 1998) which is work-intensive and requires a lot of personnel. The largest dataset of halo observations has been collected by the German "Arbeitskreis Meteore e.V. Sektion Halobeobachtungen" (AKM, https://www.meteoros.de). The community was founded in 1990 and consists of a network of about 80 volunteers who collect halo observations on a monthly basis throughout Germany, Austria, Romania and the UK. Since 1986 more than 150,000 observations of halo displays have been reported. The AKM collects information about the halo type and its duration, the type of cloud producing the halo, the weather situation during the observation (frontal system, precipitation) and more. These observations are valuable for obtaining an average frequency of the different halo displays in Europe. However, for a systematic comparison with other measurement data, continuous observations at a specific location for a long period of time are required.

An extensive long-term observation study of high-level clouds and halo displays was performed by Sassen et al. (2003), who evaluated a ∼10 year record of photographic halo observations together with measurements with a polarization lidar and other remote sensing instruments at the Facility for Atmospheric Remote Sensing (FARS) in Salt Lake City, Utah. This study is also based on visually collected halo observations. A fisheye camera, which took pictures every 20 min, was used in this study in combination with field notes and extra photographs to monitor optical displays. Sassen et al. (2003) pointed out that their optical display statistics are representative only for the observation area at FARS and that a common format for reporting atmospheric optical displays is needed to allow comparison of data from different locations. In order to perform long-term halo and cirrus observations, an automated low-maintenance system is needed which can be easily deployed at different locations.

We present the novel camera system HaloCam, designed for the automated observation of halo displays with high temporal and spatial resolution. Combined with a halo detection algorithm, HaloCam is, to our knowledge, the first fully automated camera system which can provide consistent long-term observations of halo displays. By evaluating the frequency of occurrence of halo displays and the fraction of cirrus clouds, the observations can contribute to gain more information about the dominating ice crystal properties.

The first section of this paper describes the setup and design of HaloCam. A first visual evaluation of the frequency of different halo displays using HaloCam observations is presented in Sect. 2.1. The following section explains the characterization and geometric calibration of HaloCam which is necessary for image processing and feature extraction of the halo displays. In the next section an automated halo detection algorithm based on a random forest classifier is presented and its implementation is described. Section 3.2 provides the results of the halo detection algorithm applied to HaloCam observations. Finally, the results of the halo display statistics are discussed with help of radiative transfer simulations.

## 2   The automated halo observation camera HaloCam

In order to automatically collect halo observations, the sun-tracking camera system HaloCam was developed at the Meteorological Institute (MIM) of the Ludwig-Maximilians University (LMU), Munich, and installed on the rooftop platform as

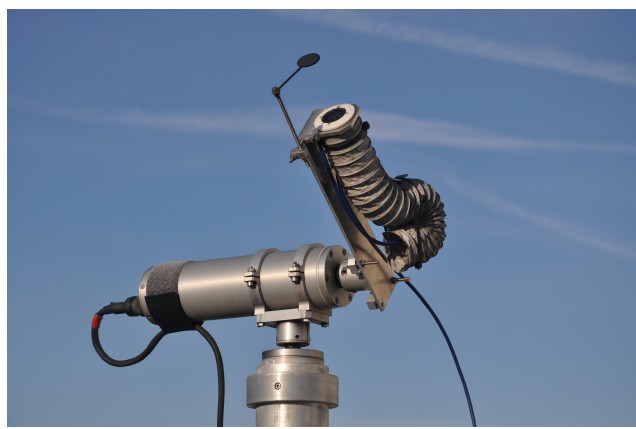

**Figure 2.** HaloCam: wide-angle camera (mobotix S14D) with circular shade on a sun-tracking mount. The mount consists of two axes with stepping motors to adjust azimuth and elevation of the camera.

shown in Fig. 2. HaloCam consists of a weather-proof wide-angle camera and is mounted on a sun-tracking system. Using a sun-tracking mount is very suitable for the observation of halo displays and later image processing since it allows to align the center of the camera with the sun. This implies that also the recorded halo displays are centered on the camera pictures. With this setup a small fixed shade is sufficient to protect the camera lens from direct solar radiation and to avoid overexposed
pixels and stray light. The mount features two stepping motors with gear boxes for adjusting the azimuth and elevation angles of the camera position as described in Seefeldner et al. (2004) with an incremental positioning of 2.16 arcmin per step. The positioning of the mount is performed by passively tracking the sun: an algorithm calculates the current position of the sun which is converted to incremental motor steps and moves the two motors accordingly. The pointing accuracy of the mount can be roughly estimated to about $\pm 0.5°$ ($2\sigma$ standard deviation) which will be explained in more detail in Sect. 2.3. The camera
(mobotix S14D) is a light-weight modular system with a RGB CMOS sensor of 1/2" size. Combined with a lens of 22 mm focal length it provides a horizontal and vertical field of view (FOV) of $90°$ and $67°$, respectively. Further specifications of the mobotix S14D camera are listed in Tab. 1. The camera is operated in an automatic exposure mode and the image region used to determine the optimum exposure time is confined to the region where the $22°$ halo occurs. This ensures that the pixels around the $22°$ halo are not saturated. The camera FOV and the sensor resolution were chosen to optimize the trade-off between a large
coverage of the sky with high spatial resolution and low image distortion. HaloCam allows to observe the $22°$ halo, sundogs, upper/lower tangent arc or circumscribed halo, which are the most frequent halo displays according to Sassen et al. (2003) and the results of the AKM.

The HaloCam observations aim at gaining a better understanding of the relationship between halo displays and typical ice crystal properties in cirrus clouds. Hence, the observations can be limited to the most frequent halo displays without loos-
ing relevant information about ice crystal shape and orientation while achieving a high spatial and temporal resolution of the scene. Every 10 s HaloCam's position relative to the sun is updated and a picture is recorded. HaloCam was installed in Sept 2013 on the rooftop platform of MIM (LMU) in Munich where operational measurements are performed by a MIRA-35 cloud

**Table 1.** HaloCam camera specifications

| Lens | |
| --- | --- |
| Equivalent 35 mm focal length | 22 mm |
| Nominal focal length | 4 mm |
| Horizontal field of view | 90° |
| Vertical field of view | 67° |

| Camera (mobotix S14D flexmount) | |
| --- | --- |
| Protection class | IP65, −30 °C to +60 °C |
| Sensor | 1/2" CMOS, RGB progressive scan |
| Sensor resolution | 3 MPixel |
| Compression formats | JPEG, MxPEG, M-JPEG |

radar (Görsdorf U. et al., 2015), a CHM15kx ceilometer (Wiegner et al., 2014) and a sunphotometer, which is part of the AERONET network (Holben et al., 1998) as well as with the institute's own sunphotometer SSARA (Toledano et al., 2009, 2011). HaloCam observations ideally complement these measurements to retrieve more detailed information about ice crystal properties.

## 2.1 HaloCam observations – a first statistical evaluation

HaloCam has been operated in Munich (Germany) since Sept 2013 where it provides continuous measurements including contributions to the ML-CIRRUS campaign in Mar/Apr 2014 (Voigt et al., 2017). Only during the ACCEPT campaign (Analysis of the Composition of Clouds with Extended Polarization Techniques, Myagkov et al. (2016)) in Oct/Nov 2014 it was installed in Cabauw (The Netherlands). A first visual evaluation of halo display frequency during ACCEPT (10 Oct until 14 Nov 2014) was performed. The results are displayed in Fig. 3 as Venn-diagram (Venn, 1880). The occurrence of each different halo type is visualized by a circle. The radius of each circle scales with the total observation time for the respective halo type. Cross sections between the circles indicate instances where two or three halo displays were visible at the same time. The observation time is given in hours. The total time of HaloCam observations, which were collected during daytime only, amounts to about 344 h. With about 30 h, halo displays were observed in almost 9% of the time. The presence of cirrus clouds within the Halo-Cam field of view was evaluated visually and amounts to about 110 h. Thus, about 27% of the cirrus clouds produced a visible halo display. The 22° halo (complete or partial) occurred in 16.2%, the sundogs in 19% and the upper tangent arcs in 7.8% of the time when cirrus clouds were present. Circumscribed halos were not observed during the campaign due to the low solar elevations. As illustrated in Fig. 3, sundogs were observed more often than 22° halos with about 21 h vs. 18 h. Thus, sundogs occurred in 70% and 22° halos in 60% of the total halo observation time (30 h). Upper tangent arcs occurred in total for about

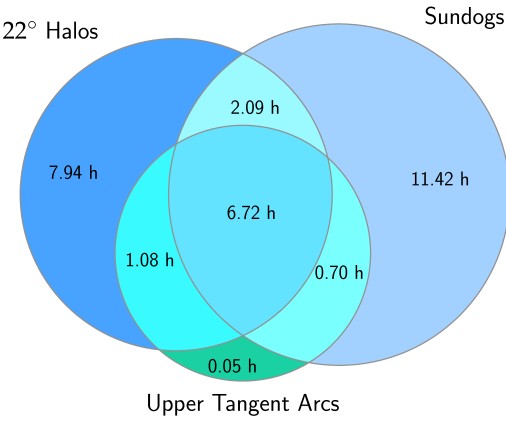

**Figure 3.** Halo display statistics from HaloCam observations during the ACCEPT campaign 10 Oct – 14 Nov 2014. The observation times of 22° halo, sundogs and upper tangent arc are provided in hours and are represented by the radii of the three circles. Cross sections of circles indicate time periods when two or three halo displays were visible simultaneously. The total observation time amounts to 344 h.

9 h (30%) and were accompanied most of the time by 22° halos and sundogs. Thus, the majority of the halo displays were produced by oriented ice crystals.

Compared to the findings of Sassen et al. (2003) the relative fraction of 22° halos is roughly similar with 50%, but sundogs with 12% and upper/lower tangent arcs with about 15% were far less frequent than observed during ACCEPT. The AKM observed the left and right sundogs with a relative frequency of 18% each, compared to 36% for the 22° halos. Although the frequency of simultaneous occurrence of the left and right sundog is unknown (from the AKM database), one can deduce that the relative frequency of sundogs is at least 18% and thus larger than the result of Sassen et al. (2003). The reasons for the differences in the observed halo frequencies could be manifold: one main reason might be that a statistical evaluation over six weeks is compared to a database of 10 (Sassen et al., 2003) and 30 years (AKM). It is possible that the observation time during ACCEPT was not long enough to yield representative results for the frequency of the different halo displays. Another factor could be the observation site. The mountains in the East of Salt Lake City, the observation site of Sassen et al. (2003), could obscure the sun during periods with low solar elevation which are favorable for the formation of sundogs. So it is possible that on average fewer sundogs could have been observed in Salt Lake City than in Cabauw which is surrounded by a rather flat landscape. Also differences in the dominating weather patterns forming cirrus clouds in Salt Lake City and Cabauw could have an impact on halo formation as discussed in Sassen et al. (2003). For the AKM and the HaloCam dataset, information about dominating weather patterns for different halo displays is not available. Furthermore, the observation period during the ACCEPT campaign from October until mid-November was dominated by low solar elevations which implies a higher chance for observing sundogs. Long-term observations have to be evaluated to obtain representative results of the frequency of the different halo types. To evaluate the large HaloCam dataset that has been collected for more than 2.5 years, an automated algorithm was developed

for the detection of 22° halos. The following sections describe how the HaloCam images are processed and which features are extracted for an automated halo detection.

## 2.2 Camera characterization and calibration

Halo displays are single scattering phenomena and thus are directly linked to the optical properties of the ice crystals producing them. The ice crystal phase function predicts the scattering angle $\Theta$ of the 22° halo relative to the sun. Thus, the analysis of the HaloCam images can be simplified significantly by mapping the image pixels to scattering angles. This means the camera has to be calibrated in order to determine the parameters for mapping the camera pixels to the real world spherical coordinate system. For this mapping the intrinsic camera parameters have to be determined, which are the focal lengths $f_x$, $f_y$ and image center coordinates $c_x$, $c_y$, as well as the distortion coefficients of the camera lens.

Different methods exist for the geometric calibration. Here, we use the method described by Zhang (2000), which is based on Heikkila and Silven (1997), to estimate the intrinsic camera parameters as well as the radial and tangential distortion parameters of the lens. This method requires several pictures of a planar pattern, for example a chessboard pattern with known dimensions, taken with different orientations. The calibration method using a chessboard pattern was implemented in OpenCV by Itseez (2015) and is described in detail by Bradski and Kaehler (2008). Using the distortion coefficients and intrinsic parameters, the camera pixels can be undistorted and mapped to the world coordinate system. Thereby a zenith ($\vartheta$) and azimuth angle ($\varphi$) relative to the image center can be assigned to each pixel. Since the image center is pointing to the center of the sun, the relative zenith angle ($\vartheta$) corresponds to the scattering angle $\Theta$ in this case.

An overlay of the scattering angle grid onto a HaloCam picture is shown in Fig. 4a with representative contour lines at $\vartheta = 22°$, 35° and 46°. From the scattering angle grid the horizontal and vertical FOV can be calculated to ~93.4° and ~70.2°, respectively. HaloCam images are recorded with a resolution of 1280×960 quadratic pixels which results in an angular resolution of ~0.07° for both the horizontal and the vertical direction. Fig. 4b shows the relative azimuth angle grid which is chosen such that the image is separated into 6 segments. For further analysis and feature extraction each of these segments is averaged azimuthally.

## 2.3 HaloCam image processing and feature extraction

For processing the HaloCam images, they can be decomposed in their red, green, and blue color channels. The brightness $I$ of each pixel, provided in digital numbers [DN], can then be represented as a data array of the size 1280×960. As an example the HaloCam image of Fig. 4 is used to demonstrate how the images are processed in case of a 22° halo. Fig. 5 shows the brightness distributions of the red, green and blue channel as a function of the scattering angle, averaged azimuthally over the uppermost image segment (no. 4 in Fig. 4b). The shaded areas around the lines in Fig. 5 represent twice the standard deviation of the averaged image region.

For analyzing the HaloCam observations several features can be extracted from the brightness distribution across the 22° halo: the scattering angle position of the brightness maximum and minimum, which are indicated in Fig. 5 by vertical dashed and dotted lines, respectively. The angular position of the 22° halo maximum ($\vartheta_{\mathrm{halo,max}}$) is found by searching for the maximum

a)

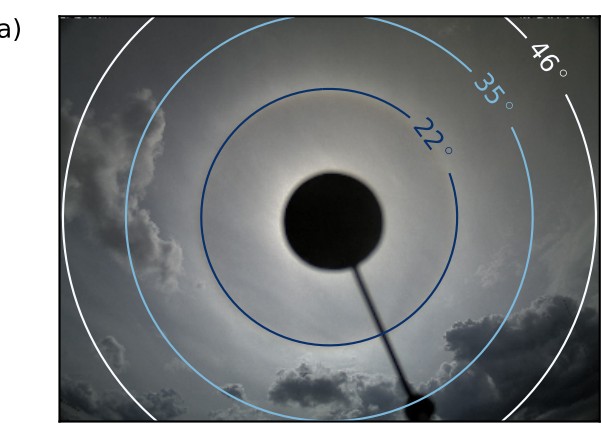

b)

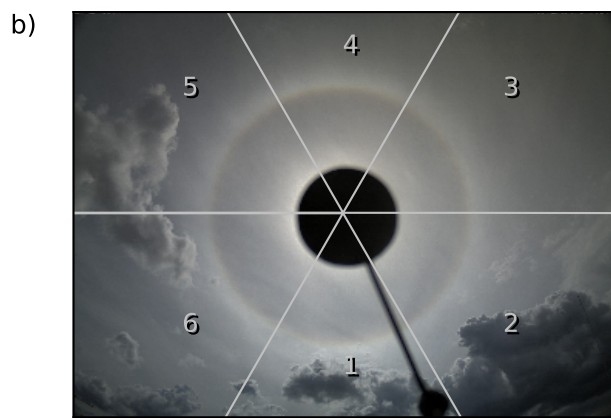

**Figure 4.** a) HaloCam image from 2014-05-12, 13:52 UTC with corresponding scattering angle ($\vartheta$) grid and representative contour lines at $22°$, $35°$ and $46°$, b) shows the relative azimuth ($\varphi$) grid with numbered labels for the 6 image segments.

brightness in the interval $(21.0°, 23.5°)$. Then the angular position of the halo minimum ($\vartheta_{\text{halo,min}}$) is determined by looking for the minimum brightness in the interval $(18.0°, \vartheta_{\text{halo,max}})$. Another important feature is the brightness contrast of the halo. In previous publications (Gayet et al., 2011; Shcherbakov, 2013; van Diedenhoven, 2014) the so-called "halo ratio" was introduced as a measure for the brightness contrast of the $22°$ and $46°$ halo in the scattering phase function. In analogy, here, the

5 halo ratio (HR) is defined as the brightness $I$ at the scattering angle of the halo maximum $\vartheta_{\text{halo,max}}$ divided by the brightness at the scattering angle of the minimum $\vartheta_{\text{halo,min}}$:

$$\text{HR} = I(\vartheta_{\text{halo,max}})/I(\vartheta_{\text{halo,min}}) \qquad (1)$$

As an example, the values for $I(\vartheta_{\text{halo,max}})$ and $I(\vartheta_{\text{halo,min}})$ are displayed in Fig. 7 by the blue triangles pointing up (max) and down (min), respectively. For clearsky conditions and homogeneous cloud cover the brightness distribution decreases from the

10 sun towards larger scattering angles, as shown in the example in Figs. 5 and 7. If $\text{HR} < 1$ the brightness at the scattering angle of the halo maximum ($I(\vartheta_{\text{halo,max}})$) is smaller than for the minimum ($I(\vartheta_{\text{halo,min}})$) which is representative for a monotonically

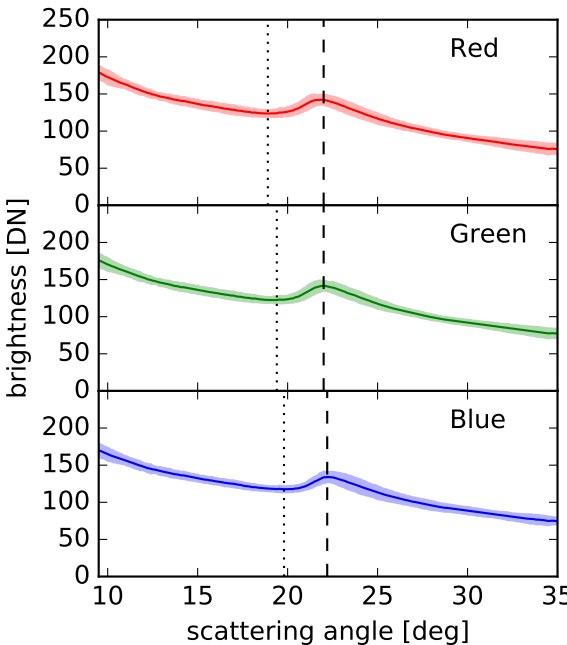

**Figure 5.** HaloCam image processing demonstrated for the measurements shown in Fig. 4, segment no. 4. The three panels show the brightness distributions (in digital numbers [DN]) for the red, green and blue image channel as a function of the scattering angle. The solid line represents the brightness averaged azimuthally over the image segment, whereas the shading indicates the $2\sigma$ standard deviation. The vertical lines pinpoint the scattering angles of the 22° halo minimum (dotted) and maximum (dashed) for the RGB channels.

**Table 2.** 22° halo features, for the example of 12 May 2014 13:52 UTC (as in Fig. 5). The relative zenith angle (which corresponds to the scattering angle) is listed for the minimum $\vartheta_{\mathrm{halo,\,min}}$ and maximum $\vartheta_{\mathrm{halo,\,max}}$ brightness of the 22° halo together with the brightness contrast, i.e. the halo ratio (HR) for the red, green and blue image channel.

|  | $\vartheta_{\mathrm{halo,\,min}}$ | $\vartheta_{\mathrm{halo,\,max}}$ | HR |
|---|---|---|---|
| Red | 18.9° | 22.0° | 1.15 |
| Green | 19.4° | 22.0° | 1.16 |
| Blue | 19.8° | 22.2° | 1.14 |

decreasing, featureless curve in this scattering angle region. This is the case for clearsky conditions or homogeneous cloud cover without halo. For $HR = 1$ the brightness at the halo maximum and minimum are the same causing a slight plateau in the brightness distribution. A distinct halo peak occurs for the condition $HR > 1$. Thus, we assume $HR = 1$ as lower threshold for the visibility of a halo. For the example of Fig. 5 the 22° halo features are compiled in Tab. 2, evaluated for the uppermost image segment. The scattering angle of the halo minimum ($\vartheta_{\mathrm{halo,\,min}}$) is smallest for the red channel and largest for the blue channel which is responsible for the reddish inner edge and the slightly blueish outer edge of the 22° halo visible in Fig. 4. It should be noted that in many cases the 22° halo appears rather white apart from a slightly reddish inner edge (Minnaert, 1937;

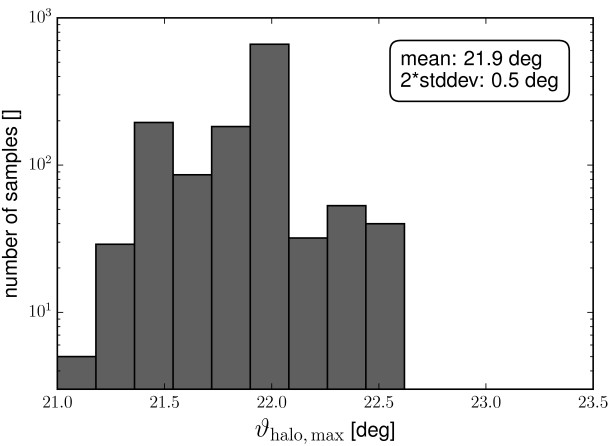

**Figure 6.** Distribution of the scattering angles of the $22°$ halo brightness maximum $\vartheta_{\mathrm{halo,\,max}}$ in degrees for 1289 randomly chosen and visually classified images using the uppermost image segment (no. 4). The mean value amounts to $21.9°$ with a $2\sigma$ confidence interval of $\pm 0.5°$. Note the logarithmic scale of the y-axis.

Vollmer, 2006). The differences between scattering angles for the three colors are smaller for $\vartheta_{\mathrm{halo,\,max}}$ with a slightly larger value for the blue channel. The halo ratio amounts to about 1.15 averaged over all three channels and is largest for the green and smallest for the blue channel.

The angular position of the $22°$ halo brightness peak ($\vartheta_{\mathrm{halo,\,max}}$) can also be used to estimate the positioning accuracy of
HaloCam relative to the sun. Fig. 6 shows a histogram of $\vartheta_{\mathrm{halo,\,max}}$ for 1289 randomly selected HaloCam pictures showing a $22°$ halo in the uppermost image segment. This segment was chosen since it contains the most pronounced halos. For a faint halo the peak in the brightness distribution is rather flat causing a larger uncertainty in finding the angular position of the peak. The mean value of $\vartheta_{\mathrm{halo,\,max}}$ amounts to $21.9°$ with a $2\sigma$-standard deviation of $0.5°$, which is a rough estimate of HaloCam's pointing accuracy. Since $\vartheta_{\mathrm{halo,\,max}}$ and $\vartheta_{\mathrm{halo,\,min}}$ are searched for within an angular interval, the pointing accuracy of $\pm 0.5°$ is
sufficient to detect the halo.

## 3   Development of an automated halo detection algorithm

The HaloCam long-term dataset from Jan 2014 until Jun 2016 was evaluated by applying a machine learning algorithm for the automated detection of halos. The algorithm was trained using features extracted from the HaloCam images. Some of these features (e.g. HR, $\vartheta_{\mathrm{halo,\,max}}$, $\vartheta_{\mathrm{halo,\,min}}$) were already described in the previous section. As a first implementation the detection
algorithm is presented here for the case of the $22°$ halo but it is possible to extend it to other halo types as well.

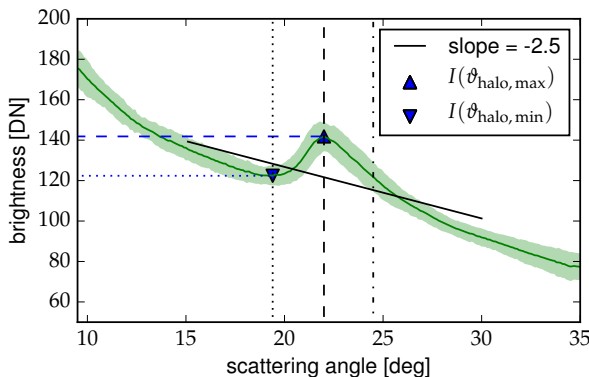

**Figure 7.** As Fig. 5 showing the first minimum (dotted) and the maximum (dashed) of the $22°$ halo for the green channel. In addition, $\vartheta_{\text{halo,end}}$ is indicated (dash-dot line) which represents the scattering angle of the same brightness as $\vartheta_{\text{halo,min}}$ and confines the halo peak. In this example $\vartheta_{\text{halo,end}}$ is located at about $24.5°$. The corresponding brightness $I(\vartheta_{\text{halo,min}})$ and $I(\vartheta_{\text{halo,max}})$ used to calculate the HR are marked with the blue triangles pointing down (min) and up (max). The regression line of the averaged brightness distribution (solid black), which is evaluated between scattering angles of $15°$ and $30°$, has a slope of -2.5 for this example.

### 3.1 Description of the classification algorithm

The detection is performed by a classification algorithm which is trained to predict whether a HaloCam picture belongs to the class "$22°$ halo" or "no $22°$ halo". For such a binary classification a decision tree can be used to create a model which predicts the class of a data sample. Details on decision trees are explained in the Appendix A. One major issue of decision trees is their
tendency to over-fitting by growing arbitrarily complex trees depending on the complexity of the data. In this study we use the random forest classifier as described by Breiman (2001), which improves the issue of over-fitting significantly by growing an ensemble of decision trees. A description of the random forest classifier used in this study is provided in the Appendix B. In principle, other classification algorithms could be used like artificial neural networks, for example. The reasons why the random forest classifier was chosen are: apart from its robustness to over-fitting it does not require much pre-processing of the
input data like scaling or normalizing. During the training of the individual trees the out-of-bag samples (i.e. the samples which were not in the trainings subsets) are used as test data and classification error estimates (e.g. out-of-bag error) can be calculated simultaneously (Breiman, 2001). In contrast to an artificial neural network, the basic structure and the internal threshold tests of the decision trees are simple to understand and can be explained by boolean logic. Henceforward, the algorithm applied to the classification of $22°$ halos will be called HaloForest.
The features used here for the classification are the $22°$ halo ratio, the scattering angle position of the halo minimum and maximum, and the scattering angle confining the halo peak $\vartheta_{\text{halo,end}}$, which are shown in Fig. 7 together with the slope of the regression line in black (solid). The halo peak is confined by $\vartheta_{\text{halo,end}}$ (dash-dotted line) which represents the scattering angle with the same brightness level as $\vartheta_{\text{halo,min}}$ in the scattering angle interval $(\vartheta_{\text{halo,max}}, 35°]$. This feature is used to ensure that

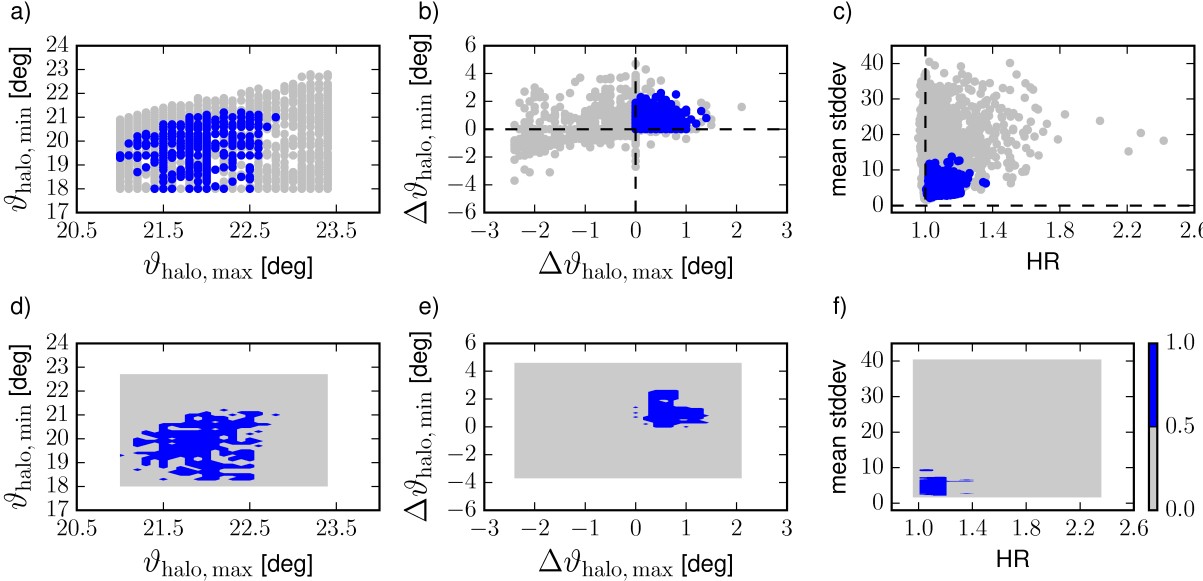

**Figure 8.** Panels a) – c): scatter plots of selected pairs of the 8 features used for training HaloForest. Training samples with(out) 22° halos are represented in blue (gray). Panels d) – f): decision boundaries of the random forest classifier for the respective feature pair. The predicted probability used for separating the classes "22° halo" (p > 0.5) and "no 22° halo" (p ≤ 0.5) is displayed in blue and gray, respectively.

the brightness for angles larger than $\vartheta_{\text{halo, max}}$ is decreasing again. The slope of the regression line serves as an estimate for the brightness gradient around the sun. For clearsky images this gradient is steeper than for overcast cases. As a measure for the colorfulness of the halo, the scattering angle difference between the blue and red channel for the halo minimum ($\Delta\vartheta_{\text{halo, min}}$) and maximum ($\Delta\vartheta_{\text{halo, max}}$) are calculated, which are defined as

$$\Delta\vartheta_{\text{halo, max}} = \vartheta_{\text{halo, max, blue}} - \vartheta_{\text{halo, max, red}}$$

5  $$\Delta\vartheta_{\text{halo, min}} = \vartheta_{\text{halo, min, blue}} - \vartheta_{\text{halo, min, red}}$$  (2)

Furthermore, the standard deviation of the brightness averaged over the image segment is used as a proxy for the inhomogeneity of the scene. These eight features are calculated for each of the six image segments separately. In order to get an impression of typical values of the training features for the two classes, Figs. 8a – c show 2-dimensional scatter plots of selected feature pairs for the upper image segment (no. 4). Features, which belong to the class "22° halo", are displayed in blue whereas the features

10  of the class "no 22° halo" are represented by gray scatter points. Fig. 8a shows the distribution of the scattering angle of the halo maximum versus minimum. The scattering angles of the halo maximum $\vartheta_{\text{halo, max}}$ are confined to a smaller interval for "22° halo" compared with "no 22° halo". However, the two classes share many data points in this projection so more features are needed to generate decision boundaries in a higher, here 8-dimensional space. Fig. 8b depicts the scattering angle difference between the blue minus the red channel for the halo maximum ($\Delta\vartheta_{\text{halo, max}}$) versus minimum ($\Delta\vartheta_{\text{halo, max}}$), which is positive

**Table 3.** Confusion matrix for HaloForest for the uppermost (no. 4) and lowermost (no. 1) image segments. The label "Predicted" refers to the class which was predicted by HaloForest whereas "True" labels the visually identified class. The true positives (correctly classified "22° halo") are printed in bold font. False positives ("no 22° halo" classified as "22° halo") and false negatives are listed on the other diagonal. The results are provided with a $2\sigma$ standard deviation.

|  | | **Predicted** | |
|---|---|---|---|
|  | Segment 4: | 22° halo | no 22° halo |
| **True** | 22° halo | **97.3 ± 1.9 %** | 0.4 ± 0.3 % |
|  | no 22° halo | 2.7 ± 0.9 % | **99.6 ± 0.2 %** |
|  | | | |
|  | Segment 1: | 22° halo | no 22° halo |
| **True** | 22° halo | **88.5 ± 7.1 %** | 0.5 ± 0.5 % |
|  | no 22° halo | 11.5 ± 3.5 % | **99.5 ± 0.2 %** |

for the "22° halo" class since the inner edge (smaller $\vartheta$) of the 22° halo is slightly red. The HR, which is shown in Fig. 8c, takes values between 1 and ∼1.3 for "22° halos". Images with a low mean standard deviation of the image segment indicate rather homogeneous scenes which are present most of the time when a 22° halo is visible. Figs. 8a – c visualize that the two classes "22° halo" and "no 22° halo" can not be separated easily since the values of the features overlap. The lower panels of

Fig. 8d – f display the regions which are detected as "22° halo" (blue) and "no 22° halo" (gray) by the trained algorithm.

For each of the six image segments an individual classifier was trained using a dataset of visually classified HaloCam images which were chosen randomly from the dataset. The performance of the classifiers was tested using a random selection of 30% of the dataset which was excluded from training. This procedure was repeated 100 times to get statistically significant results for the performance of the classifier. Tab. 3 shows the confusion matrix for the classifier of the segments directly above

(no. 4) and below the sun (no. 1) which represent the two extreme cases of the performance of the six different classifiers: the upper part of the 22° halo has a higher brightness contrast compared to the lower part which is often obstructed by the horizon. For the training of HaloForest 1289 samples with a 22° halo and 5181 samples without 22° halo were used for the uppermost segment (no. 4). The lowermost segment (no. 1) was trained with 296 and 3370 samples of the classes 22° halo and no 22° halo, respectively. The lines of the confusion matrix indicate the true class labels of the samples ("22° halo" and "no

22° halo"), whereas the columns contain the predicted class labels. The number of true positive and negative (in bold) as well as false positive and negative classifications are evaluated and provided with a $2\sigma$ standard deviation. The correct classification of "22° halo" is maximum for the uppermost image segment (no. 4) with about 98% and minimum for the lowermost segment with about 89%. The correct classification of "no 22° halo" is overall higher than 99%, so the HaloForest algorithm seems to be able to separate the two classes well. The performance of the other four segments ranges between the results of the upper

and lowermost segments.

**Table 4.** Confusion matrix as in Tab. 3 for 470 randomly selected HaloCam images between Jan 2014 – Jun 2016, evaluated for segments 3, 4, and 5.

|  |  | Predicted | |
| --- | --- | --- | --- |
|  |  | 22° halo | no 22° halo |
| **True** | 22° halo | **88.8 %** | 2.8 % |
|  | no 22° halo | 11.2 % | **97.2 %** |

## 3.2 Application of the halo detection algorithm

HaloForest is used to evaluate the dataset HaloCam has collected in Munich between Jan 2014 and Jun 2016. To ensure a high classification accuracy, only the classifiers for the upper image segments (3, 4, and 5) were used (cf. Tab. 3). A HaloCam image was assigned to the class "22° halo" if at least one of the image segments 3, 4, or 5 predicts a 22° halo. Applying a proba-
bility threshold of $p > 0.5$, 22° halos were detected in 152 h. Relative to the total observation time during daylight of 7345 h, 22° halos occurred in about 2.1% of the time. As an additional test, the classification accuracy of HaloForest was checked for 470 randomly chosen HaloCam images for the "22° halo" and "no 22° halo" class within this long-term observation period in Munich. The confusion matrix for this test is provided in Tab. 4 for the image segments no. 3, 4, and 5 together. More than 88% of the 22° halos are classified correctly and less than 12% are classified incorrectly as 22° halos.
Images were incorrectly classified as 22° halo predominantly due to small bright clouds or contrails in a blue sky, or structures in overcast conditions which happen to cause a peak in the averaged brightness distribution at a scattering angle of 22°.
Based on these results we investigated the fraction of cirrus clouds which produced a halo in Munich during this time period. The total frequency of occurrence of cirrus clouds was determined by independent data of co-located CHM15kx ceilometer observations (Wiegner and Geiß, 2012). To guarantee consistent observational conditions, only ceilometer measurements in
the absence of low-level clouds were considered. Proprietary software of the ceilometer automatically provides up to three cloud base heights with a temporal resolution of 15 s. The detection is based on the fact that in case of clouds backscatter signals are significantly larger than the background noise. The sensitivity of the ceilometer is sufficient to even detect clouds near the tropopause during daytime. Since ceilometers, however, do not provide depolarization information, the discrimination between water and ice clouds was made by means of the cloud base temperature $T_{base}$. Sassen and Campbell (2001) state
that cirrus cloud base temperatures ranged between $-30$ °C and $-40$ °C during the 10-year observation period at the FARS observation site. As a temperature threshold is not an unambiguous criterion for the existence of ice clouds, we have calculated the frequency of occurrence for three different temperatures: $-20$ °C, $-30$ °C, and $-40$ °C. If $T_{base}$ is lower than the given temperature threshold, the cloud is considered a "cirrus cloud". The temperature profiles were obtained from routine radiosonde ascents of the German Weather Service at Oberschleißheim (WMO station code 10868), which is located about 13 km north
of the HaloCam site. During the time period from Jan 2014 until Jun 2016 a fraction of 5.6% cirrus clouds was detected for a cloud base temperature of $T_{base} < -20$ °C. Towards lower cloud base temperatures the amount of detected cirrus clouds

decreases to 3.5% for $T_{base} < -30\ ^\circ C$ and 1.9% for $T_{base} < -40\ ^\circ C$.

Due to the different pointing directions of the ceilometer (towards zenith) and HaloCam (towards sun) the instruments observe different regions of the sky. This is accounted for by pre-screening the data for 1-h time intervals when the ceilometer detected a cirrus cloud, subject to data availability for both instruments. The subsequent analysis of cirrus fraction and halo frequency of occurrence is based on the full temporal resolution of 15 s and 10 s, respectively. Relative to the amount of detected cirrus clouds about 25% occurred together with a 22° halo for the image segments 3, 4, and 5. This fraction does not change much for the different cloud base temperatures (26.4% for $T_{base} < -20\ ^\circ C$ and 24.5% for $T_{base} < -40\ ^\circ C$) since the fraction of detected clouds decreases together with the detected halos for lower temperatures. According to the confusion matrix in Tab. 4, 88.8% of the detected "22° halos" are real halos, while 2.8% of the "no 22° halos" are actually "22° halos". Correcting the result for the estimated false classifications, the fraction of "halo-producing" cirrus clouds amounts to about $25\% \cdot 88.8\% + 75\% \cdot 2.8\% \approx 24\%$. The comparison of the ceilometer and HaloCam data implies that about 25% of the cirrus clouds contain some fraction of smooth, hexagonal ice crystals. Sassen et al. (2003) observed a fraction of 37.3% cirrus clouds which produced a 22° halo within 1-hour time intervals. The results most likely differ because the observations originate from different locations which might be dominated by different mechanisms for cirrus formation. It has to be noted however, that the evaluation method is very sensitive to the sampling strategy of the observations: the fraction of "halo-producing" cirrus clouds increases to more than 50%, if the HaloCam observations are binned to 1-hour intervals, which are counted as containing a halo regardless of their duration.

For comparison, the fraction of cirrus clouds producing a halo display was evaluated visually for the HaloCam observations during the ACCEPT campaign and amounts to about 27% including 22° halos, sundogs and upper/lower tangent arcs (cf. Section 2.1). This value is also lower than the result provided by Sassen et al. (2003) who observed any of the three halo types in about 54% of the 1-hour periods with cirrus.

The current version of HaloForest discriminates only between the two classes "22° halo" and "no 22° halo". Thus, interference with other halo types as sundogs or upper/lower tangent arcs and circumscribed halos might occur at certain solar elevations. The position of sundogs relative to the sun depends on the solar zenith angle (SZA) and can be calculated analytically as described in Wegener (1925); Tricker (1970); Minnaert (1993); Liou and Yang (2016). The sundogs are located at scattering angles close to the 22° halo for large SZAs and occur at larger scattering angles for small SZAs, i.e. high solar elevations. Fig. 9 shows the same HaloCam image with the azimuth segments as Fig. 4b. In addition, the minimum scattering angle of the sundogs are calculated as a function of the SZA and represented by the red and green squares. The SZAs range between 90° and 35° with a resolution of 1°. The two white circles centered around the sun at scattering angles of 21.0° and 23.5° indicate the mask which is used to find the scattering angle of the 22° halo peak. For SZA $\leq 67°$ the sundog positions are located outside this mask and cannot be mis-classified as 22° halo (green squares). The red squares represent sundog positions which are located within this mask and might therefore be mis-classified. This is the case for SZAs between 90° and 67°. To obtain an estimate of the fraction of sundogs which are mis-classified as 22° halo 1000 randomly selected HaloCam images were counter-checked visually. It revealed that only 6 images showing sundogs without 22° halo in the segments (3–5) were

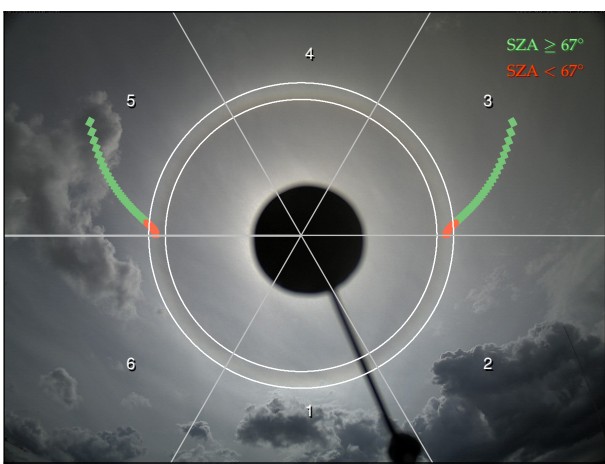

**Figure 9.** HaloCam image as in Fig. 4b. The red and green squares indicate the minimum scattering angle of the sundogs as a function of the solar zenith angle (SZA). The SZA ranges between $90°$ and $35°$ with $1°$ resolution. The mask used to search for the $22°$ halo peak is displayed by the two white circles and covers scattering angles between $21.0°$ and $23.5°$. Sundog positions located within this mask might be mis-classified as $22°$ halo and are marked as red. These positions correspond with SZAs between $90°$ and $67°$. For smaller SZAs (higher solar elevations) the sundogs are located outside the mask and cannot be mis-classified as $22°$ halo by the algorithm.

mis-classified as $22°$ halo, which is $< 1\%$. Upper tangent arcs could be detected by the uppermost image segment (no. 4) and might be mis-classified as $22°$ halo. For very small SZAs (high solar elevations) the tangent arcs merge to form the circumscribed halo which could be detected in the segments 3 and 5 as well. The same procedure was repeated for these halo types: 1000 randomly selected images were checked for the presence of tangent arcs and circumscribed halos without $22°$ halo yielding 28 images or 2.8%. However, if only a fragment of a halo is visible in the uppermost segment, it is generally difficult to discriminate between an upper tangent arc or circumscribed halo and a $22°$ halo.

The halo classification algorithm was presented for $22°$ halos, but it is possible to include training data for other halo types as well. With the current version of HaloForest and the co-located ceilometer observations the fraction of cirrus clouds producing a halo display was estimated to about 25% for Munich between Jan 2014 and Sept 2016. Extending HaloForest for the detection of other halo types, as sundogs for example, the fraction of "halo-producing" cirrus clouds could easily exceed 25%. In principle, HaloCam could also be equipped with a wide-angle lens to observe halo displays in a larger region of the sky, however at the expense of the spatial resolution.

## 4 Sensitivity study of the visibility of the $22°$ halo and interpretation of halo statistics

In this section we discuss the factors that contribute to the visibility of halo displays using the example of the $22°$ halo. This is important for a more detailed interpretation of the fraction of "halo-producing" cirrus clouds and ice crystal roughness.

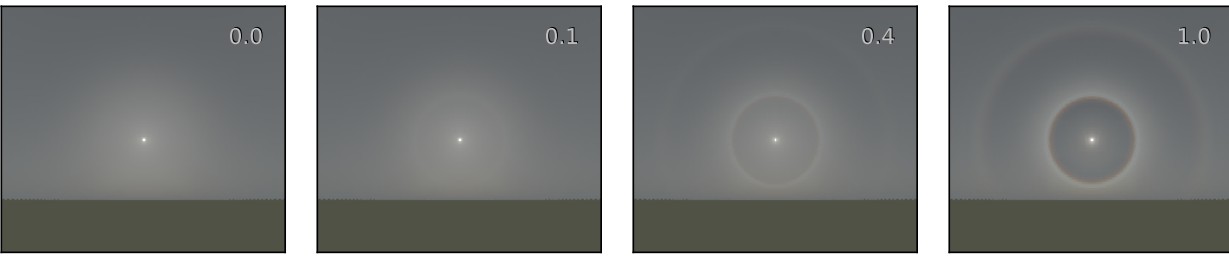

**Figure 10.** Sky radiance simulations with libRadtran (Mayer and Kylling, 2005) using the DISORT solver for a solar zenith angle of $60°$, a viewing azimuth angle range of $0° – 160°$ and for viewing zenith angles from $10° – 110°$ (i.e. from the zenith to $20°$ below the horizon). The simulations were performed for a spectral range of $380 – 780$ nm (5 nm steps), weighted with the spectral sensitivity of the human eye. A homogeneous cirrus cloud layer with optical thickness of 1 was assumed. Solid column ice crystal optical properties of Yang et al. (2013) with an effective radius of $80\,\mu m$ were used. Aerosol scattering was not considered. The four panels show radiative transfer simulations with different fractions of smooth solid columns ranging from 0% to 100%, as indicated by the labels. A background of severely roughened solid columns is assumed with fractions changing from 100% to 0%, accordingly.

The effect of varying cloud optical thickness on the visibility of halo displays was already investigated by Kokhanovsky (2008); Gedzelman and Vollmer (2008); Gedzelman (2008) using radiative transfer simulations. Kokhanovsky (2008) performed simulations of the brightness contrast of the $22°$ halo as a function of the cirrus optical thickness using the radiative transfer model SCIATRAN neglecting molecular and aerosol scattering. The results show a linear decrease of the halo contrast with increasing optical thickness. Gedzelman (2008) and Gedzelman and Vollmer (2008) used the model HALOSKY for radiative transfer simulations of halos with varying cloud optical thickness. HALOSKY considers single scattering by air molecules, aerosol particles and cloud particles assuming homogeneous, plane-parallel atmospheric layers. Multiple scattering is calculated only within the cloud by a Monte Carlo subroutine. Gedzelman and Vollmer (2008) show results for radiance simulations of the $22°$ halo in the principal plane below and above the sun. They found that the radiance at the bottom of the halo reaches a maximum value for smaller COT ($\approx 0.25$) than the radiance at the top of the cloud ($\approx 0.63$).

In this study, radiative transfer simulations were performed using the libRadtran radiative transfer package (Mayer and Kylling, 2005; Emde et al., 2016) and the DISORT ("discrete ordinate technique") solver (Stamnes et al., 1988; Buras et al., 2011). LibRadtran allows for an accurate simulation of Rayleigh scattering, molecular absorption, aerosols, surface albedo as well as water and ice clouds. DISORT is a one-dimensional solver regarding the atmosphere as a number of homogeneous, plane-parallel layers. Radiative transfer simulations of a cirrus cloud were performed assuming a homogeneous ice cloud layer with optical thickness 1 (at 550 nm) at a height between $10 – 11$ km. Fig. 10 shows simulations using different fractions of smooth solid columns (0%, 10%, 40%, 100%) and assuming a background of severely roughened solid columns. All ice crystals have an effective radius of $80\,\mu m$. The optical properties were chosen from the database by Yang et al. (2013). The sun is located at a zenith angle of $60°$. Sky radiance was calculated for an angular range between $0° – 160°$ in the azimuth direction and $10° – 110°$ (i.e. from $10°$ off-zenith to $20°$ below the horizon) in the zenith direction, which corresponds to

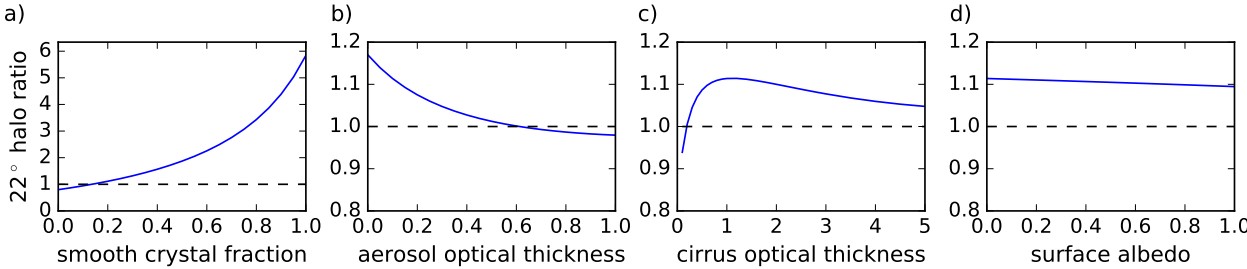

**Figure 11.** Sensitivity studies of the 22° halo ratio at 550 nm (as defined in Eq. 1) as a function of smooth crystal fraction, aerosol optical thickness (AOT), cirrus optical thickness (COT), and surface albedo (from left to right). The radiative transfer simulations were performed with libRadtran assuming an ice cloud between $10-11$ km using ice crystal optical properties as in Fig. 10 for a solar zenith angle of $60°$. The dashed line indicates $HR = 1$, which marks the threshold for the visibility of a halo display. The default parameters, i.e. if not varied, are 20% smooth solid columns, $AOT = 0.2$, $COT = 1.0$, and $\mathrm{albedo} = 0.0$.

the view of a wide-angle camera. The simulations were performed for a spectral range of $380-780$ nm (5 nm steps) and the results were weighted with the spectral sensitivity of the human eye according to CIE 1986, as implemented in specrend (http://www.fourmilab.ch/documents/specrend/).

Aerosol scattering was not considered and a spectral surface albedo of grass was chosen (Feister and Grewe, 1995). For 0%
(first panel of Fig. 10) all ice crystals are rough and thus no 22° or 46° halo is visible. For a fraction of 10% smooth crystals the 22° halo starts to form which is in agreement with the findings of van Diedenhoven (2014). The 46° halo becomes visible for a fraction of 40% smooth crystals. For 100% smooth crystals both 22° and 46° halo reach a maximum brightness contrast for the respective cirrus optical thickness.

Fig. 11 depicts the sensitivity of the halo brightness contrast, represented by the halo ratio as a function of the smooth ice
crystal fraction (a), the aerosol optical thickness (b), the cirrus optical thickness (c) and the surface albedo (d) for a wavelength of 550 nm. As in Fig. 10 a SZA of $60°$ was chosen and the ice cloud was defined between $10-11$ km. The halo ratio was determined in the principal plane above the sun. The dashed lines indicate a halo ratio of 1, which we defined as threshold for the visibility of halo displays. Fig. 11a shows clearly that for a smooth crystal fraction of $>10\%$ the halo ratio exceeds 1 and the 22° halo is visible. An increasing aerosol optical thickness causes a decrease of the HR, which is displayed in Fig. 11b.
For a typical value of $AOT = 0.2$ the HR is reduced by $\sim 10\%$ compared to an aerosol free atmosphere. Fig. 11c shows how the HR is determined by the optical thickness of the cirrus cloud (COT) itself. We observe a maximum value for COT$\sim 1$. For a very thin cirrus Rayleigh and aerosol scattering become dominant resulting in a small HR. Only when COT is larger than the optical thickness of the background (here Rayleigh and aerosol), the HR approaches its maximum value. For large COT, multiple scattering reduces the contrast of the halo feature and the HR decreases, similar to the findings of Kokhanovsky
(2008). However, as Gedzelman and Vollmer (2008) point out, the halo peak might still be visible up to an optical thickness of $\sim 5$ due to the pronounced maximum in the scattering phase function.

A higher surface albedo causes longer photon paths through the atmosphere and thus a higher chance of multiple scattering

(Fig. 11d). Reflected photons therefore cause a higher "background" brightness. It is evident that a brighter background causes a weaker brightness contrast of the halo display. In general, the effect of the surface albedo on the HR is small compared with the effect of AOT or COT. Halo displays are a geometric optics phenomenon, which means that they emerge only when the particle size is much larger than the wavelength (Fraser, 1979; Mishchenko and Macke, 1999; Garrett et al., 2007; Flatau

and Draine, 2014) which also depends on the aspect ratio of the crystals (Um and McFarquhar, 2015). The solar zenith angle (SZA) affects the halo brightness contrast indirectly by increasing the photon path length through the atmosphere for large SZAs and thus increasing the amount of multiple scattering (not shown). This effect is the same for different viewing zenith angles which explains the fact why the 22° halo is always brightest at the top (directly above the sun) and faintest below the sun.

With this knowledge we can now discuss further implications of the fraction of "halo-producing" cirrus clouds. HaloCam observations showed that ∼25% of the cirrus clouds, which were visible from the ground, produced a 22° halo. It can be argued that these cirrus clouds contained a certain amount of smooth, hexagonal ice crystals. By analyzing ice crystal single scattering properties van Diedenhoven (2014) showed that a minimum fraction of 10% smooth hexagonal ice crystal columns is sufficient to produce a 22° halo. In case of ice crystal plates the minimum fraction of smooth crystals for a visible halo is

much larger with about 40%. Thus, if the exact ice crystal habits of the cirrus cloud are unknown, which is typically the case, the minimum amount of smooth ice crystals probably lies in a range of 10% to 40%. This implies that even for a large fraction of irregular or small ice crystals a halo might still be visible. A larger fraction of smooth ice crystals, however, could well be possible for halos with larger HR, i.e. increased brightness contrast. Multiple scattering of the cirrus cloud or atmosphere was not considered by van Diedenhoven (2014). This study revealed that during the ∼2.5 years of HaloCam observations in

Munich about 75% of the cirrus clouds did not produce a 22° halo. For favorable atmospheric conditions, i.e. COT∼1 and negligible aerosol scattering, the maximum fraction of rough ice crystals ranges between 60% and 90%. Thus, it is possible that the majority of cirrus clouds during the observation period in Munich contain a large fraction of rough ice crystals. This would support the hypothesis of e.g. Knap et al. (2005); Baran and Labonnote (2006); Baran et al. (2015) who found that on average rough ice crystals better reproduce remote sensing radiance measurements than assuming crystals with smooth surface.

However, if multiple scattering by cirrus clouds or aerosol is accounted for, the minimum fraction of smooth crystals could be much larger in the case of "halo-producing" cirrus clouds. The actual fraction of smooth ice crystals for cirrus clouds with visible halo display must be analyzed in detail and will be addressed in future work. This requires HaloCam observations to be complemented by radiative transfer simulations and additional measurements of aerosol and cirrus optical thickness. These additional measurements can be provided by radar, lidar and sunphotometer measurements available at the observation site at

MIM, LMU in Munich. Surface albedo measurements can be obtained from satellite data products.

## 5 Summary and Conclusions

In this paper we present HaloCam, a novel sun-tracking camera system for the automated observation of halo displays. The camera has a field of view of 90° in the horizontal and 67° in the vertical direction and a resolution of 1280×960 quadratic

pixels which yields an angular resolution of 0.07°. The camera system records images in RGB color space and JPEG compression every 10 s. It automatically tracks the sun so that the halo displays stay centered relative to the camera. HaloCam observations can contribute to a better understanding of ice crystal shape, surface roughness and orientation by long-term observations of halo displays. Different halo displays are caused by different ice crystal shapes and orientations. The most frequent halo displays are formed by either randomly oriented or oriented plates and columns and therefore contain the most important information about ice crystal properties. Therefore, the camera setup was optimized for observing 22° halos, sundogs and upper/lower tangent arcs or circumscribed halos with high spatial and temporal resolution without loosing relevant information.

An initial visual evaluation of the frequency of halo displays reveals that for the 6-weeks ACCEPT campaign sundogs were observed more often than 22° halos. Together with the observations of upper tangent arcs this implies that about 73% of the observed halo displays were caused by oriented ice crystals. This result differs from the findings of other studies, like Sassen et al. (2003), who observed that 22° halos are more frequent than sundogs and upper tangent arcs based on a dataset of about 10 years. A visual evaluation of the presence of cirrus clouds during the campaign showed that about 27% produced a 22° halo, sundogs, or upper/lower tangent arcs. Sassen et al. (2003) found that in about 54% of the 1-hour cirrus periods any of the three halo types was visible. It should be highlighted that the evaluation method is very sensitive to the sampling method and the temporal resolution of the observations.

For evaluating the long-term HaloCam observations in Munich an automated halo detection algorithm, called HaloForest, was developed. HaloForest is presented here for the detection of 22° halos but it can be extended for the detection of other halo types such as sundogs and upper/lower tangent arcs. The algorithm is based on a random forest classifier and was trained and tested against visually evaluated observations. With more than 88% of the test samples correctly classified as "22° halos" and more than 97% correctly classified as "no 22° halo", HaloForest is able to separate the two classes well. Applied to the more than 2.5 years of data, HaloForest detected 22° halos in about 2% of the total observation time during daylight.

A first estimate of ice crystal roughness was performed by evaluating the frequency of cirrus clouds that were accompanied by halo displays. For the long-term halo observations in Munich, co-located ceilometer measurements were used to evaluate the fraction of cirrus clouds. About 25% of the detected cirrus clouds in Munich occurred together with a 22° halo. Extending HaloForest for more halo types (e.g. sundogs) would increase the fraction of "halo-producing" cirrus clouds above 25%.

These results imply that the majority of cirrus clouds which did not produce a visible halo, very likely, contained primarily rough ice crystals and 25% (or 27% for ACCEPT) of the clouds contained at least a certain fraction of smooth, hexagonal ice crystals. Based on the study by van Diedenhoven (2014) a minimum fraction of smooth crystals of 10% in case of columns or 40% in case of plates can be estimated for the halo-producing cirrus clouds if multiple scattering and scattering by aerosol is neglected. These assumptions allow to determine a minimum fraction of smooth crystals in halo-producing cirrus clouds. If multiple scattering by cloud and aerosol is accounted for, the required fraction of smooth ice crystals could be significantly larger than 40%. To further constrain the fraction of rough ice crystals, more detailed quantitative studies are needed which will be addressed in future work. This analysis requires radiative transfer simulations and additional constraints which can be provided by radar, lidar and sunphotometer measurements available at the observation site at LMU in Munich.

This study highlights the potential and feasibility of a completely automated method to collect and evaluate halo observations. These long-term observations allow to estimate the average fraction of rough ice crystals in cirrus clouds. Quantitative evaluation of halo radiance distributions can contribute to systematically investigate ice crystal surface roughness, shape and orientation in cirrus clouds. Implemented on different sites, HaloCam in combination with the HaloForest detection algorithm can provide a consistent dataset for climatological studies.

## Appendix A: Decision Trees

The subsequent sections provide more details on decision trees and the random forest classifier presented in Sect. 3.
The following description is based on Alpaydin (2010) and Raschka (2015). Decision trees start with a root node followed by internal decision nodes, branches and terminal nodes, called leaves. A typical example of a single decision tree, as used

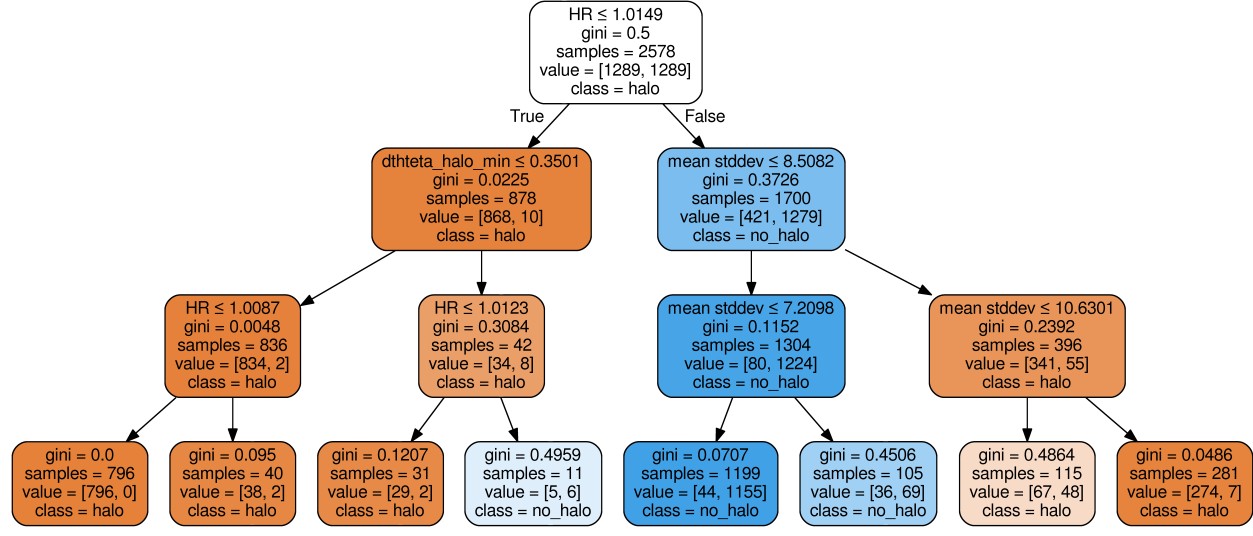

**Figure 12.** Example for a decision tree for a selection of three HaloCam image features confined to a maximum depth of three layers. The two classes, "halo" and "no halo" are depicted by red and blue color. The transparency of the color represents the impurity of the class.

for HaloForest, is shown in Fig. 12. For a better visualization, the tree is grown using only three of the eight features and is pruned to a depth of three layers. The explanation provided here focuses on the structure of tree rather than the exact numbers of the threshold tests which differ from the ones used by HaloForest. The halo ratio (HR), the mean standard deviation, and $\Delta\vartheta_{\mathrm{halo,\,min}}$ are used as features in this case, which are displayed in the first line of each node box with the respective threshold test. At each decision node a threshold test is applied to one element of the $n$-dimensional feature vector (here, $n = 3$) which best splits the set of samples. The metric to determine the best split in this study is the Gini impurity index, which is defined

by Raschka (2015) as

$$I_G(t) = 1 - \sum_{i=1}^{c} p(i|t)^2 \tag{A1}$$

with $c$ the number of classes and $p(i|t)$ the fraction of samples which belongs to class $i$ at node $t$. The Gini index takes a minimum value for the maximum information gain (all the samples at node $t$ belong to one class) and the index is maximum for a uniform distribution. The discrete result (here, True or False) of the threshold test decides which of the following branches is chosen. The node boxes are connected by arrows representing the branches of the tree. They are colored depending on the dominating class in the samples which is noted at the bottom of each box: red for "22° halo" and blue for "no 22° halo". The more transparent the color the higher the impurity of the classes and the larger the Gini impurity index. This splitting process is repeated recursively at each child node until a leaf node is reached. A leaf node is hit when all the samples in the subset belong to the same class, or when splitting does not add more information. By repeating this recursive decision process the $n$-dimensional feature space is subdivided into the pre-defined classes on a path following from the root down. Fig. 8 shows examples of the resulting decision boundaries as 2-dimensional projections for a selection of feature pairs. The decision tree is trained using a set of labeled training samples. During training the tree grows by adding branches and leaves depending on the complexity of the data, which can lead to over-fitting. By growing an ensemble of decision trees this issue can be improved, which is the idea of random forest classifiers.

## Appendix B:  Random Forest Classifier Implementation

In this study we use the random forest classifier, which is described by Breiman (2001) and implemented in the python module scikit-learn (Pedregosa et al. (2011), version 0.18.1). The trees are trained by applying the bootstrap aggregation (bagging) method (Breiman, 1996), i.e. by using a subset of the training samples which is chosen randomly with replacement and has the same size as the original input samples. This implementation predicts the class of a sample by averaging the probabilistic prediction of all individual decision trees instead of using the majority vote among the trees. The function call allows to define a number of parameters: the number of trees is set to 100 and a maximum number of 3 features ($\log_2(n)$ with $n$ features) is considered for searching the best split. These parameters are chosen to minimize the out-of-bag (OOB) error, as shown in Fig. 13. For an increasing number of estimators (trees) the OOB error stabilizes for around 100 trees and is in general smaller for a confined number of features considered at each split.

*Acknowledgements.*  The radiosonde data were downloaded from http://weather.uwyo.edu/upperair/sounding.html of the University of Wyoming, College of Engineering, Department of Atmospheric Science. The halo observations during the ACCEPT campaign research received funding by the European Union Seventh Framework Program (FP7/2007-2013) under grant agreement n° 262254. We thank Markus Garhammer (LMU, Munich) and Marc Allaart (KNMI, The Netherlands) for their support during the campaign.

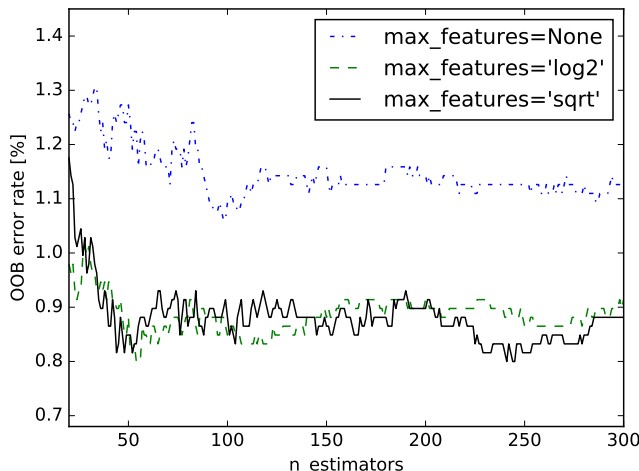

**Figure 13.** Out-of-bag error for different values of n_estimators (number of trees) for three different realizations of the random forest classifier by changing the number of features considered at each split.

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
