# Peer review of "Ice Crystal Characterization in Cirrus Clouds: A Sun-tracking Camera System and Automated Detection Algorithm for Halo Displays"

_Atmospheric Measurement Techniques, 2017_

## Referee Comment (RC1) · Anonymous Referee #1 · 19 Mar 2017

This manuscript reports on the observations of 22-degree halo associated with cirrus clouds by using a sun-tracking system called HaloCam. An automated halo detection algorithm was clearly explained. A brief history of halo observations was reviewed. The HaloCam observations during six weeks were analyzed. The findings were compared with the counterparts revealed by two other datasets, the FARS (Facility for Atmospheric Remote Sensing) and AKM (Arbeitskreis Meteore e.v. Sketion Halobeobachtungen) analyses. Furthermore, theoretical simulations of halo with four combinations of smooth and rough ice crystals were performed.

Overall, the manuscript is well organized and clearly written. No major technical errors were noted. The manuscript in its present form can be essentially accepted for publication as is.

Typographical errors:

Line 3 on page 3: "150.000" should be "150,000".

Line 17 on page 5: "Also AKM state" should be "Also AKM states". Or, should it be better to state "In addition, the AKM observations reveal"

Suggestions for future studies:

1. From the upper panel of Fig.4, the HaloCam system is capable of observing 46-degree halo. As correctly pointed out in the manuscript, the ratio of 22-degree halo to 46-degree halo contains rich information about ice crystal aspect ratio. Thus, it is suggested that the present study based on HaloCam be extended to analyses of 46-degree halo. 2. Ice crystals in the form of individual bullets or bullet rosettes have been extensively assumed (based on some in-situ microphysical property observations). These ice crystals produce a halo at approximately 10 degrees. Did the HaloCam system ever observe this type of halos?

---

## Referee Comment (RC2) · B. van Diedenhoven (Referee) · 30 Mar 2017

This paper introduces a new instrument and method for automated detection of ice cloud halo displays. The detection of a halo indicates the presence of smooth crystals in ice clouds. Such smooth crystals generally have a larger scattering asymmetry parameter than roughened crystals and thus studies on the presence of smooth crystals are relevant for constraining cirrus optical properties. Statistics on the presence of smooth crystals are scarce and the presented instrument seems very useful to improve such statistics when deployed around the globe.

The paper is very well written and relevant for publication in Atmospheric Measurement Techniques. I do have some suggestions to possibly improve the paper.

1. A visual inspection of the halo images is presented on page 5. Please indicate whether partial 22-degree halos are also counted in the statistics. Also, in my experience sundogs often appear much brighter than 22 halos, and therefor may be more easily detected. Could that have skewed the statistics?

2. The automated detection algorithm focuses on 22-degree halos. Could you please discuss whether and how the algorithm might be biased by the presence of other optical phenomena such as sundogs and tangent arcs? I can imagine that in case of a 22-degree additional sundogs present halo change the angular width of features seen in segments 3 and 5, which might lead to a false negative detection. On the other hand, the presence of a sundog without a 22-degree halo might lead to a false positive detection of a 22-degree halo.

3. Related to the discussion of influence of optical thickness of the visibility of halos, it would be good to include the follow paper: Kokhanovsky, A.: The contrast and brightness of halos in crystalline clouds, Atm. Res., 89, 110–112, doi:10.1016/j.atmosres.2007.12.006, 2008.

4. On page 17, percentages of the fraction of rough particles are estimated. Since these are based on the minimum percentage of smooth crystals needed for halo features, it seems to me that the deduced fraction of rough particles are maximum values. That is, a lower percentage of rough particles would of course also produce a halo, and probably a brighter one.

Minor corrections:

Line 13, page 15: Remove "are" from the final sentence.

Line 1, page 15: I suggest to refer to section 2.1 here

СЗ

---

## Referee Comment (RC3) · Anonymous Referee #3 · 5 Apr 2017

The comment was uploaded in the form of a supplement: http://www.atmos-meas-tech-discuss.net/amt-2017-17/amt-2017-17-RC3-supplement.pdf

---

## Referee Comment (RC4) · B. van Diedenhoven (Referee) · 5 Apr 2017

This is in reply to the preface included in the review by reviewer #3. I feel the remarks are largely inspired by my open review of this paper and I would therefor like to take the time to respond.

I regularly review papers for the Copernicus journals as an official reviewer (as in this case). I agree with the reviewer that the system is a bit odd at times and at least different from the usual review process. I also agree that one's review could be biased by previous reviews that are posted online and I try to refrain from reading previous

reviews before writing my own. Actually, I think the idea proposed by the reviewer to have the open discussion as a second step after a more traditional review process is one to be considered by Copernicus.

Whenever I am asked by a Copernicus editor to review a paper, I usually do so anonymously. In this case, I chose not to stay anonymous in order to show my appreciation for a young scientist who delivered an excellent paper in my opinion. In my own experience as an author publishing in Copernicus journals, I appreciated reviewers adding their names to constructive reviews of my papers. I do not think there are many 'rewards' to be gained by adding your name to a review. On the other hand, in cases such as this, I do not think it can do any harm either.

Sincerely,

Bastiaan van Diedenhoven (Reviewer #2)

———————————————————

---

## Author Comment (AC2) · 24 May 2017

The reply to the review report of referee #2 is provided as supplement.

Please also note the supplement to this comment:
http://www.atmos-meas-tech-discuss.net/amt-2017-17/amt-2017-17-AC2-supplement.pdf

---

## Author Comment (AC1)

**Reply to comments by referee #1**

We thank the referee for carefully reviewing the manuscript and for the valuable suggestions and comments.

Remark: The referee's comments are highlighted in blue. Figure numbers in the authors' reply refer to the figures in the original manuscript. New/changed figures are included at the end of the document. Snippets included in the revised manuscript are highlighted by an additional indent and quotation marks.

**Line 3 on page 3: "150.000" should be "150,000".**

Changed.

Line 17 on page 5: "Also AKM state" should be "Also AKM states". Or, should it be better to state "In addition, the AKM observations reveal".

Thank you for the hint, we changed the sentence:

"The AKM observed the left and right sundogs with a relative frequency of 18% each, compared to 36% for the  $22^{\circ}$  halos."

Suggestions for future studies: 1. From the upper panel of Fig.4, the HaloCam system is capable of observing 46-degree halo. As correctly pointed out in the manuscript, the ratio of 22-degree halo to 46-degree halo contains rich information about ice crystal aspect ratio. Thus, it is suggested that the present study based on HaloCam be extended to analyses of 46-degree halo.

Thank you for pointing this out. Indeed observations of the  $46^{\circ}$  halo in addition to the  $22^{\circ}$  halo would further increase the information content on ice crystal properties. However, with this setup of HaloCam the  $46^{\circ}$  halo is very close to the edge of the image which makes an evaluation difficult. Currently we are testing another observation setup with the camera tilted upward so that the upper part of the  $22^{\circ}$  halo and the  $46^{\circ}$  halo are located inside the image. For changes see answer below.

2. Ice crystals in the form of individual bullets or bullet rosettes have been extensively assumed (based on some in-situ microphysical property observations). These ice crystals produce a halo at approximately 10 degrees. Did the HaloCam system ever observe this type of halos?

With this setup of HaloCam it is not possible to observe a  $10^{\circ}$  halo since this viewing angle is just covered by the circular shade. To observe this type of halo the distance between shade and camera could be increased (the size should remain the same to shield the whole lens from direct sunlight). Depending on the dynamic range of the camera the simultaneous observation of the  $10^{\circ}$  halo together with the  $22^{\circ}$  or even the  $46^{\circ}$  halo on the same image (i.e. with the same exposure time) could be difficult. Since referee #3 raised a similar question, we included the following sentence:

"In principle, HaloCam could also be equipped with a wide-angle lens to observe halo displays in a larger region of the sky, however at the expense of the spatial resolution."

---

## Author Comment (AC3)

**Reply to comments by referee #3**

We thank the referee for carefully reviewing the manuscript and for the valuable suggestions and comments.

Remark: The referee's comments are highlighted in blue. Figure numbers in the authors' reply refer to the figures in the original manuscript. New/changed figures are included at the end of the document. Snippets included in the revised manuscript are highlighted by an additional indent and quotation marks.

Page 1 line 2: when mentioning sundogs you should first mention the scientific description parhelia and then maybe the non-technical term sundog.

Thank you for the hint, we changed this accordingly.

"Further frequently observed halo displays are the parhelia of the 22° halo, commonly called sundogs, which are caused by sunlight refracted by horizontally oriented hexagonal plates."

Page 1, lines 4-8: As I understand the system was mostly in Munich but also for a 6 week period in the NL. In the text, the Cabauw period is mentioned once, but later on it is always stated that the measurements in Munich are discussed from 01/2014 to 06/2016. I propose to explicitly state which periods were used and which excluded for the Munich measurements.

We added the specific time of the ACCEPT campaign in the abstract and clarified the text.

"An initial visual evaluation of the frequency of halo displays for the ACCEPT (Analysis of the Composition of Clouds with Extended Polarization Techniques) field campaign from October to mid November 2014 showed that sundogs were observed more often than  $22^{\circ}$  halos."

Page2, top: I miss other refs. For example you use some old ones, but why not also Pernter Exners excellent book. Concerning general refs: you only mention Tape94. Tape has later written another interesting book on halos as well: Atmospheric halos and the search for angle X, W. Tape, J. Moilanen, AGU (Am Geophys. Union) 2006. Also, it is rather odd that you refer to the famous book by Minnaert with the year 1993. This is just a new translation of the much older NL book with the first edition dating back to 1937. Newcomers to the field might assume Minnaert is a contemporary scientist, please correct ref. by adding e.g. the original information.

Thank you for pointing this out. We added/changed the respective references in the manuscript.

Page 2, line 7: Probably you assume that everybody already knows about the 22° halo with respect to the circumscribed halo / tangent arcs. My personal experience is that most people just know about the 22° halo and have problems in understanding the differences to the other ones. Maybe just give short explanations in one or two sentences describing the differences (sun elevation). And it also seems to me that you use tangent arcs and circumscribed halo synonymously, if so: please make a respective statement somewhere

We added a short explanation in the introduction. We did not intent to use tangent arcs and circumscribed halos synonymously and clarified in the manuscript where necessary.

"Hexagonal ice crystal columns with their long axis oriented horizontally form another halo type: the upper and lower tangent arcs. Their shape changes with the solar elevation. When the sun is close to the zenith both the upper and lower tangent arc merge to the circumscribed halo."

Page 3, line 32: maybe clarify .... This implies that the most important recorded (?) halo...

We changed the sentence accordingly:

"This implies that also the recorded halo displays are centered on the camera pictures."

Page 3, line 6: I miss the main results about halo frequencies in Germany from AKM. Later on you give similar results from Sassen in the US, you discuss the Munich results in Bavaria, so why not add results reported from all over Germany as well? Meteorological conditions in Germany should be closer to yours than the ones in the US.

The results of AKM are given on page 5, line 17/18 (in the discussion manuscript) and compared with the HaloCam observations and the results of Sassen. We pointed out that "AKM observed the left and right sundogs with a relative frequency of 18% each compared to 36% for the 22° halos. Although the frequency of simultaneous occurrence of the left and right sundog is unknown (from the AKM database), one can deduce that the relative frequency is at least 18% and is thus larger than the result of Sassen et al. (2003)."

"For the AKM and the HaloCam dataset, information about dominating weather patterns for different halo displays is not available."

Page 4 line 9: here you suddenly also mention tangent arcs, previously only  $22^{\circ}$  halo and circumscribed halo (see above). As mentioned above, briefly discuss all relevant halo features which may be observable with your equipment and then explain why you mainly focus on  $22^{\circ}$  halos...

We added a description of upper/lower tangent arcs and circumscribed halos to the introduction (cf. reply to previous comment).

"Hexagonal ice crystal columns with their long axis oriented horizontally form another halo type: the upper and lower tangent arcs. Their shape changes with the solar elevation. When the sun is close to the zenith both the upper and lower tangent arc merge to the circumscribed halo."

... and that / under which conditions your data may contain misinterpreted circumscribed halos / tangent arcs.

Referee #2 raised a similar question. Therefore, we added a discussion whether tangent arcs, circumscribed halos and sundogs could be mis-classified as  $22^{\circ}$  halo and how HaloForest could be extended to separate these other halo types from the  $22^{\circ}$  halo.

"The current version of HaloForest discriminates only between the two classes "22° halo" and "no 22° halo". Thus, interference with other halo types as sundogs or upper/lower tangent arcs and circumscribed halos might occur at certain solar elevations. The position of sundogs relative to the sun depends on the solar zenith angle (SZA) and can be calculated analytically as described in Wegener (1925); Tricker (1970); Minnaert (1993); Liou and Yang (2016). The sundogs are located at scattering angles close to the  $22^{\circ}$  halo for large SZAs and occur at larger scattering angles for small SZAs, i.e. high solar elevations. Fig. 1 (now Fig. 9 in the manuscript) shows the same HaloCam image with the azimuth segments as Fig. 4b. In addition, the minimum scattering angle of the sundogs are calculated as a function of the SZA and represented by the red and green squares. The SZAs range between  $90^{\circ}$ and  $35^{\circ}$  with a resolution of  $1^{\circ}$ . The two white circles centered around the sun at scattering angles of  $21.0^{\circ}$  and 23.5° indicate the mask which is used to find the scattering angle of the 22° halo peak. For SZA  $\leq 67^{\circ}$  the sundog positions are located outside this mask and cannot be mis-classified as  $22^{\circ}$  halo (green squares). The red squares represent sundog positions which are located within this mask and might therefore be mis-classified. This is the case for SZAs between 90° and  $67^{\circ}$ . To obtain an estimate of the fraction of sundogs which are mis-classified as 22° halo 1000 randomly selected HaloCam images were counter-checked visually. It revealed that only 6 images showing sundogs without 22° halo in the segments (3–5) were mis-classified as 22° halo, which is

"In analogy, here, the halo ratio (HR) is defined as the brightness I at the scattering angle of the halo maximum  $\vartheta_{\text{halo,max}}$  divided by the brightness at the scattering angle of the minimum  $\vartheta_{\text{halo,min}}$ :

$$HR = I(\vartheta_{halo, max}) / I(\vartheta_{halo, min})$$
(1)

As an example, the values for  $I(\vartheta_{halo, max})$  and  $I(\vartheta_{halo, min})$  are displayed in Fig. 2 by the blue triangles pointing up (max) and down (min), respectively. For clearsky conditions and homogeneous cloud cover the brightness

distribution decreases from the sun towards larger scattering angles, as shown in the example in Figs. ?? and 2. If HR < 1 the brightness at the scattering angle of the halo maximum  $(I(\vartheta_{halo, max}))$  is smaller than for the minimum  $(I(\vartheta_{halo, min}))$  which is representative for a monotonically decreasing, featureless curve in this scattering angle region. This is the case for clearsky conditions or homogeneous cloud cover without halo. For HR = 1 the brightness at the halo maximum and minimum are the same causing a slight plateau in the brightness distribution. A distinct halo peak occurs for the condition HR > 1. Thus, we assume HR = 1 as lower threshold for the visibility of a halo."

Page 16, line 14: you mention that in multiple scattering HR decreases. This is plausible, but please also give reference. Or did you only intend to give a qualitative plausibility statement?

The simulations intend only to show qualitatively the effect of multiple scattering on the visibility of the  $22^{\circ}$  halo. Kokhanovsky (2008) shows similar findings. We included this reference.

"For large COT, multiple scattering reduces the contrast of the halo feature and the HR decreases, similar to the findings of Kokhanovsky (2008). However, as Gedzelman and Vollmer (2008) point out, the halo peak might still be visible up to an optical thickness of  $\sim 5$  due to the pronounced maximum in the scattering phase function."

Sect. 5 Conclusions: I miss somewhere – not necessarily here – a discussion about potential reasons for the observed differences in halo frequency observations Sassen/AKM/your work. For example discuss meteorological differences in cloud formation due to different climates etc. You only mentioned a little bit on pages 5,6.

We included Sassen et al. (2003) as reference for the discussion of the meteorological conditions during the presence of the halo displays. However, the correlation between the occurrence of halo displays and certain weather patterns have not been evaluated for the HaloCam observations in Munich and have also not been analyzed for the AKM dataset. The main reason for the differences between our observations and Sassen/AKM is the small temporal range of the observations during the ACCEPT campaing of 6 weeks compared to the long-term observations of Sassen (10 years) and AKM (30 years). We added the following sentences:

"Also differences in the dominating weather patterns forming the cirrus clouds in Salt Lake City and Cabauw could have an impact on halo formation as discussed in Sassen et al. (2003). For the AKM and the HaloCam dataset, information about dominating weather patterns for different halo displays is not available."

I assume – if extending the halo algorithm to parhelia, the color separation may be an additional criterion for distinguishing halo types.

We use information about the color separation already for the detection of  $22^{\circ}$  halos ( $\Delta \vartheta_{halo, min/max}$ ). We agree that this might be even more important for detecting parhelia since they have a more distinct separation of colors than the  $22^{\circ}$  halos.

Outlook: maybe say a few words what may in principle be possible. Can you imagine using a wide angle lens and also detect the colorful circumzenithal arcs which also have a reasonably high observation frequency in Germany?

Using a lens with a wider field of view allows to observe more halo displays of course at the expense of a reduced spatial resolution. See also response to referee #1. We added the following text to the manuscript:

"In principle, HaloCam could also be equipped with a wide-angle lens to observe halo displays in a larger region of the sky, however at the expense of the spatial resolution."

Fig. 11 discussing your decision tree: In the paper you mention and show figures with HR around 1.1 to 1.15, but your criteria in the tree give numbers much lower (1.01 or so). Please comment. You may want to discuss what typical brightness variations under daylight conditions are considered to be easily perceived by the human eye.

Regarding Fig. 11 we state that the tree uses only three of the eight features and is pruned to a depth of three layers. So the criteria used in this case are not the same as used by HaloForest. In general, the halo ratios could well be lower than 1.1 and 1.15 as shown in Fig. 8. When comparing the halo ratios derived from pictures with jpeg compression it should also be kept in mind that the brightness on these images does not change linearly but is compressed by a gamma correction. So the halo ratios could slightly deviate from the brightness contrast in the "raw" signal measured by the camera sensor. We added the following sentence:

"A typical example of a single decision tree, as used for HaloForest, is shown in Fig. 11. For a better visualization, the tree is grown using only three of the eight features and is pruned to a depth of three layers. The explanation provided here focuses on the structure of tree rather than the exact numbers of the threshold tests which differ from the ones used by HaloForest."

**Figure 1.** HaloCam image as in Fig. 4b. The red and green squares indicate the position of the sundogs as a function of the solar zenith angle (SZA). The SZA ranges between 90° and 35° with 1° resolution. The mask used to search for the 22° halo peak is displayed by the two white circles and covers scattering angles between  $21.0^{\circ}$  and  $23.5^{\circ}$ . Sundog positions located within this mask might be mis-classified as  $22^{\circ}$  halo and are marked as red. These positions correspond with SZAs between 90° and 67°. For smaller SZAs (higher solar elevations) the sundogs are located outside the mask and cannot be mis-classified as  $22^{\circ}$  halo by the algorithm.

**Figure 2.** As Fig. 5 showing the first minimum (dotted) and the maximum (dashed) of the 22° halo for the green channel. In addition,  $\vartheta_{\text{halo, end}}$  is indicated (dash-dot line) which represents the scattering angle of the same brightness as  $\vartheta_{\text{halo, min}}$  and confines the halo peak. In this example  $\vartheta_{\text{halo, end}}$  is located at about 24.5°. The corresponding brightness  $I(\vartheta_{\text{halo, min}})$  and  $I(\vartheta_{\text{halo, max}})$  used to calculate the HR are marked with the blue triangles pointing down (min) and up (max). The regression line of the averaged brightness distribution (solid black), which is evaluated between scattering angles of 15° and 30°, has a slope of -2.5 for this example.

**References**

Gedzelman, S. D.: Simulating halos and coronas in their atmospheric environment, Appl. Opt., 47, H157–H166, doi:10.1364/AO.47.00H157, 2008.

Gedzelman, S. D. and Vollmer, M.: Atmospheric Optical Phenomena and Radiative Transfer, Bulletin of the American Meteorological Society, 89, 471–485, doi:10.1175/BAMS-89-4-471, http://dx.doi.org/10.1175/BAMS-89-4-471, 2008.

Kokhanovsky, A.: The contrast and brightness of halos in crystalline clouds, Atmos. Res., 89, 110–112, doi:10.1016/j.atmosres.2007.12.006, 2008.

---

## Author Response (AR1)

**Reply to comments by referee #1**

We thank the referee for carefully reviewing the manuscript and for the valuable suggestions and comments.

Remark: The referee's comments are highlighted in blue. Figure numbers in the authors' reply refer to the figures in the original manuscript. New/changed figures are included at the end of the document. Snippets included in the revised manuscript are highlighted by an additional indent and quotation marks.

Line 3 on page 3: "150.000" should be "150,000".

> Changed.

Line 17 on page 5: "Also AKM state" should be "Also AKM states". Or, should it be better to state "In addition, the AKM observations reveal".

> Thank you for the hint, we changed the sentence:
>
>> "The AKM observed the left and right sundogs with a relative frequency of 18% each, compared to 36% for the 22° halos."

Suggestions for future studies: 1. From the upper panel of Fig.4, the HaloCam system is capable of observing 46-degree halo. As correctly pointed out in the manuscript, the ratio of 22-degree halo to 46-degree halo contains rich information about ice crystal aspect ratio. Thus, it is suggested that the present study based on HaloCam be extended to analyses of 46-degree halo.

> Thank you for pointing this out. Indeed observations of the 46° halo in addition to the 22° halo would further increase the information content on ice crystal properties. However, with this setup of HaloCam the 46° halo is very close to the edge of the image which makes an evaluation difficult. Currently we are testing another observation setup with the camera tilted upward so that the upper part of the 22° halo and the 46° halo are located inside the image. For changes see answer below.

2. Ice crystals in the form of individual bullets or bullet rosettes have been extensively assumed (based on some in-situ microphysical property observations). These ice crystals produce a halo at approximately 10 degrees. Did the HaloCam system ever observe this type of halos?

> With this setup of HaloCam it is not possible to observe a 10° halo since this viewing angle is just covered by the circular shade. To observe this type of halo the distance between shade and camera could be increased (the size should remain the same to shield the whole lens from direct sunlight). Depending on the dynamic range of the camera the simultaneous observation of the 10° halo together with the 22° or even the 46° halo on the same image (i.e. with the same exposure time) could be difficult. Since referee #3 raised a similar question, we included the following sentence:
>
>> "In principle, HaloCam could also be equipped with a wide-angle lens to observe halo displays in a larger region of the sky, however at the expense of the spatial resolution."

**Reply to comments by referee #2**

We thank the referee for carefully reviewing the manuscript and for the valuable suggestions and comments.

Remark: The referee's comments are highlighted in blue. Figure numbers in the authors' reply refer to the figures in the original manuscript. New/changed figures are included at the end of the document. Snippets included in the revised manuscript are highlighted by an additional indent and quotation marks.

1. A visual inspection of the halo images is presented on page 5. Please indicate whether partial 22-degree halos are also counted in the statistics. Also, in my experience sundogs often appear much brighter than 22 halos, and therefor may be more easily detected. Could that have skewed the statistics?

The 22° halo observations include both complete and partial halos, the latter occurring more often than the complete 22° halo similar to the observations of Sassen et al. (2003). We added this information for clarification. We agree that bright and colorful sundogs could be more easily detected by eye than a faint 22° halo. However, the 22° halo covers a larger area on the HaloCam images and might therefore compensate for this. Also, looking at a sequence of images helps to detect stationary features within moving clouds even if the brightness contrast is low. In general, a visual halo detection (on images or in the sky) is never unbiased and might vary between different observers.

2. The automated detection algorithm focuses on 22-degree halos. Could you please discuss whether and how the algorithm might be biased by the presence of other optical phenomena such as sundogs and tangent arcs? I can imagine that in case of a 22-degree additional sundogs present halo change the angular width of features seen in segments 3 and 5, which might lead to a false negative detection. On the other hand, the presence of a sundog without a 22-degree halo might lead to a false positive detection of a 22-degree halo.

You are raising an interesting question. We added a paragraph to the paper to discuss possible mis-classifications due to other halo types (see also comment of referee #3):

"The current version of HaloForest discriminates only between the two classes "22° halo" and "no 22° halo". Thus, interference with other halo types as sundogs or upper/lower tangent arcs and circumscribed halos might occur at certain solar elevations. The position of sundogs relative to the sun depends on the solar zenith angle (SZA) and can be calculated analytically as described in Wegener (1925); Tricker (1970); Minnaert (1993); Liou and Yang (2016). The sundogs are located at scattering angles close to the 22° halo for large SZAs and occur at larger scattering angles for small SZAs, i.e. high solar elevations. Fig. 1 (now Fig. 9 in the manuscript) shows the same HaloCam image with the azimuth segments as Fig. 4b. In addition, the minimum scattering angle of the sundogs are calculated as a function of the SZA and represented by the red and green squares. The SZAs range between 90° and 35° with a resolution of 1°. The two white circles centered around the sun at scattering angles of 21.0° and 23.5° indicate the mask which is used to find the scattering angle of the 22° halo peak. For SZA $\leq 67°$ the sundog positions are located outside this mask and cannot be mis-classified as 22° halo (green squares). The red squares represent sundog positions which are located within this mask and might therefore be mis-classified. This is the case for SZAs between 90° and 67°. To obtain an estimate of the fraction of sundogs which are mis-classified as 22° halo 1000 randomly selected HaloCam images were counter-checked visually. It revealed that only 6 images showing sundogs without 22° halo in the segments (3–5) were mis-classified as 22° halo, which is $< 1\%$. Upper tangent arcs could be detected by the uppermost image segment (no. 4) and might be mis-classified as 22° halo. For very small SZAs (high solar elevations) the tangent arcs merge to form the circumscribed halo which could be detected in the segments 3 and 5 as well. The same procedure was repeated for these halo types: 1000 randomly selected images were checked for the presence of tangent arcs and circumscribed halos without 22° halo yielding 28 images or 2.8%. However, if only a fragment of a halo is visible in the uppermost segment, it is generally difficult to discriminate between an upper tangent arc or circumscribed halo and a 22° halo."

3. Related to the discussion of influence of optical thickness of the visibility of halos, it would be good to include the follow paper: Kokhanovsky, A.: The contrast and brightness of halos in crystalline clouds, Atm. Res., 89, 110-112, doi:10.1016/j.atmosres.2007.12.006, 2008.

Thank you for this hint, we included the reference together with Gedzelman and Vollmer (2008); Gedzelman (2008), suggested by referee #3, and briefly described their results.

"The effect of varying cloud optical thickness on the visibility of halo displays was already investigated by Kokhanovsky (2008); Gedzelman and Vollmer (2008); Gedzelman (2008) using radiative transfer simulations. Kokhanovsky (2008) performed simulations of the brightness contrast of the 22° halo as a function of the cirrus optical thickness using the radiative transfer model SCIATRAN neglecting molecular and aerosol scattering. The results show a linear decrease of the halo contrast with increasing optical thickness. Gedzelman (2008) and Gedzelman and Vollmer (2008) used the model HALOSKY for radiative transfer simulations of halos with varying cloud optical thickness. HALOSKY considers single scattering by air molecules, aerosol particles and cloud particles assuming homogeneous, plane-parallel atmospheric layers. Multiple scattering is calculated only within the cloud by a Monte Carlo subroutine. Gedzelman and Vollmer (2008) show results for radiance simulations of the 22° halo in the principal plane below and above the sun. They found that the radiance at the bottom of the halo reaches a maximum value for smaller COT ($\approx 0.25$) than the radiance at the top of the cloud ($\approx 0.63$)."

4. On page 17, percentages of the fraction of rough particles are estimated. Since these are based on the minimum percentage of smooth crystals needed for halo features, it seems to me that the deduced fraction of rough particles are maximum values. That is, a lower percentage of rough particles would of course also produce a halo, and probably a brighter one.

Thank you for pointing this out. This is exactly what we aimed to conclude from the observations. We adapted this paragraph to highlight that the derived percentages for rough ice crystals are a maximum value.

[revised manuscript text omitted]

**Reply to comments by referee #3**

We thank the referee for carefully reviewing the manuscript and for the valuable suggestions and comments.

Remark: The referee's comments are highlighted in blue. Figure numbers in the authors' reply refer to the figures in the original manuscript. New/changed figures are included at the end of the document. Snippets included in the revised manuscript are highlighted by an additional indent and quotation marks.

Page 1 line 2: when mentioning sundogs you should first mention the scientific description parhelia and then maybe the non-technical term sundog.

Thank you for the hint, we changed this accordingly.

"Further frequently observed halo displays are the parhelia of the 22° halo, commonly called sundogs, which are caused by sunlight refracted by horizontally oriented hexagonal plates."

Page 1, lines 4-8: As I understand the system was mostly in Munich but also for a 6 week period in the NL. In the text, the Cabauw period is mentioned once, but later on it is always stated that the measurements in Munich are discussed from 01/2014 to 06/2016. I propose to explicitly state which periods were used and which excluded for the Munich measurements.

We added the specific time of the ACCEPT campaign in the abstract and clarified the text.

"An initial visual evaluation of the frequency of halo displays for the ACCEPT (Analysis of the Composition of Clouds with Extended Polarization Techniques) field campaign from October to mid-November 2014 showed that sundogs were observed more often than 22° halos."

Page2, top: I miss other refs. For example you use some old ones, but why not also Pernter Exners excellent book. Concerning general refs: you only mention Tape94. Tape has later written another interesting book on halos as well: Atmospheric halos and the search for angle X, W. Tape, J. Moilanen, AGU (Am Geophys. Union) 2006. Also, it is rather odd that you refer to the famous book by Minnaert with the year 1993. This is just a new translation of the much older NL book with the first edition dating back to 1937. Newcomers to the field might assume Minnaert is a contemporary scientist, please correct ref. by adding e.g. the original information.

Thank you for pointing this out. We added/changed the respective references in the manuscript.

Page 2, line 7: Probably you assume that everybody already knows about the 22° halo with respect to the circumscribed halo / tangent arcs. My personal experience is that most people just know about the 22°halo and have problems in understanding the differences to the other ones. Maybe just give short explanations in one or two sentences describing the differences (sun elevation). And it also seems to me that you use tangent arcs and circumscribed halo synonymously, if so: please make a respective statement somewhere

We added a short explanation in the introduction. We did not intent to use tangent arcs and circumscribed halos synonymously and clarified in the manuscript where necessary.

"Hexagonal ice crystal columns with their long axis oriented horizontally form another halo type: the upper and lower tangent arcs. Their shape changes with the solar elevation. When the sun is close to the zenith both the upper and lower tangent arc merge to the circumscribed halo."

Page 3, line 32: maybe clarify .... This implies that the most important recorded (?) halo...

We changed the sentence accordingly:

"This implies that also the recorded halo displays are centered on the camera pictures."

Page 3, line 6: I miss the main results about halo frequencies in Germany from AKM. Later on you give similar results from Sassen in the US, you discuss the Munich results in Bavaria, so why not add results reported from all over Germany as well? Meteorological conditions in Germany should be closer to yours than the ones in the US.

The results of AKM are given on page 5, line 17/18 (in the discussion manuscript) and compared with the HaloCam observations and the results of Sassen. We pointed out that "AKM observed the left and right sundogs with a relative frequency of 18% each compared to 36% for the 22° halos. Although the frequency of simultaneous occurrence of the left and right sundog is unknown (from the AKM database), one can deduce that the relative frequency is at least 18% and is thus larger than the result of Sassen et al. (2003)."

"For the AKM and the HaloCam dataset, information about dominating weather patterns for different halo displays is not available."

Page 4 line 9: here you suddenly also mention tangent arcs, previously only 22° halo and circumscribed halo (see above). As mentioned above, briefly discuss all relevant halo features which may be observable with your equipment and then explain why you mainly focus on 22° halos...

We added a description of upper/lower tangent arcs and circumscribed halos to the introduction (cf. reply to previous comment).

"Hexagonal ice crystal columns with their long axis oriented horizontally form another halo type: the upper and lower tangent arcs. Their shape changes with the solar elevation. When the sun is close to the zenith both the upper and lower tangent arc merge to the circumscribed halo."

... and that / under which conditions your data may contain misinterpreted circumscribed halos / tangent arcs.

Referee #2 raised a similar question. Therefore, we added a discussion whether tangent arcs, circumscribed halos and sundogs could be mis-classified as 22° halo and how HaloForest could be extended to separate these other halo types from the 22° halo.

"The current version of HaloForest discriminates only between the two classes "22° halo" and "no 22° halo". Thus, interference with other halo types as sundogs or upper/lower tangent arcs and circumscribed halos might occur at certain solar elevations. The position of sundogs relative to the sun depends on the solar zenith angle (SZA) and can be calculated analytically as described in Wegener (1925); Tricker (1970); Minnaert (1993); Liou and Yang (2016). The sundogs are located at scattering angles close to the 22° halo for large SZAs and occur at larger scattering angles for small SZAs, i.e. high solar elevations. Fig. 1 (now Fig. 9 in the manuscript) shows the same HaloCam image with the azimuth segments as Fig. 4b. In addition, the minimum scattering angle of the sundogs are calculated as a function of the SZA and represented by the red and green squares. The SZAs range between 90° and 35° with a resolution of 1°. The two white circles centered around the sun at scattering angles of 21.0° and 23.5° indicate the mask which is used to find the scattering angle of the 22° halo peak. For SZA $\leq 67°$ the sundog positions are located outside this mask and cannot be mis-classified as 22° halo (green squares). The red squares represent sundog positions which are located within this mask and might therefore be mis-classified. This is the case for SZAs between 90° and 67°. To obtain an estimate of the fraction of sundogs which are mis-classified as 22° halo 1000 randomly selected HaloCam images were counter-checked visually. It revealed that only 6 images showing sundogs without 22° halo in the segments (3–5) were mis-classified as 22° halo, which is $< 1\%$. Upper tangent arcs could be detected by the uppermost image segment (no. 4) and might be mis-classified as 22° halo. For very small SZAs (high solar elevations) the tangent arcs merge to form the circumscribed halo which could be detected in the segments 3 and 5 as well. The same procedure was repeated for these halo types: 1000 randomly selected images were checked for the presence of tangent arcs and circumscribed halos without 22° halo yielding 28 images or 2.8%. However, if only a fragment of a halo is visible in the uppermost segment, it is generally difficult to discriminate between an upper tangent arc or circumscribed halo and a 22° halo."

Page 4 line 10: you mention Sassen's US results. Are the German AKM results consistent, i.e. do you expect the same order for the frequency of different observed halo types?

The German AKM results are similar to the results of Sassen et al. (2003). Both references state that 22° halos, sundogs and upper/lower tangent arcs are the most frequent halo displays. We added this reference.

"HaloCam allows to observe the 22° halo, sundogs, upper and lower tangent arc, which are the most frequent halo displays according to Sassen et al. (2003) and the results of the AKM."

Page 5 line 11: 30/110=0.27 is definitely closer to 1/4 than to the mentioned 1/3.

This is correct. We changed the values to 27% throughout the manuscript.

Page 7, line 9: I would add quadratic when describing the pixels.

Done.

Page 8, Fig. 5: I miss information of how you made sure that your halo pixels were not saturated. Or did you exclude images with saturated pixels?

You are addressing an important point. We did not explicitly filter out saturated pixels. However, we confined the region for the automatic exposure adjustment to the region where the 22° halo occurs to ensure that the exposure time is optimized to the region of interest. We do not expect that saturated pixels are an issue for the presented dataset as we did not encounter pictures with overexposed pixels in the 22° halo region when compiling the training data set or inspecting images for testing the classification algorithm. We added a sentence to the manuscript for clarification.

"The camera is operated in an automatic exposure mode and the image region used to determine the optimum exposure time is confined to the region where the 22° halo occurs. This ensures that the pixels around the 22° halo are not saturated."

Page 8 Sect. 2.3: I wonder whether you also found some odd radius halos. If so, your choice of angular intervals may not always work. Or did you also test for smaller angular features with different intervals? And do you have a plot frequency of halo observations versus halo angles?

For this study we focused only on the 22° halo. In principle it would be possible to chose different angular intervals to discriminate between odd radius halos. This would be interesting for a future study. Fig. 6 in the manuscript shows the frequency of 22° halo observations as a function of the scattering angle of the halo maximum. So far we did not encounter indications of odd-radius halos.

Table 2: I miss a more thorough discussion, for example based on your own simulations. Are these experimental results in agreement with your theoretical expectations?

The values provided in Tab. 2 should only provide an example of the angular position of the 22° halo features resulting from the described method used to analyze the HaloCam images. Since the results are shown for the uppermost segment only, a small shift in the angles due to a mis-alignment of the camera might be possible. We adapted the explanation and added references.

"The scattering angle of the halo minimum ($\vartheta_{\mathrm{halo,\,min}}$) is smallest for the red channel and largest for the blue channel which is responsible for the reddish inner edge and the slightly blueish outer edge of the 22° halo visible in Fig. 4b. It should be noted that in many cases the 22° halo appears rather white apart from a slightly reddish inner edge (Minnaert, 1937; Vollmer, 2006)."

Table 2: Of course the red inner edge was expected. Did you observe this always? Table 2 only refers to a single halo.

The example in Figs. 4, 5 and Tab. 2 shows a bright and colorful 22° halo. We also observed 22° halos with less pronounced separation of colors which appeared rather white. Most images, however show a discernible reddish inner edge. Fig. 8b) shows a scatter plot of the difference between the scattering angles of halo maximum and minimum which are used as a measure of how colorful the halo appeared. We adapted the text in the manuscript as stated in the reply to the previous comment.

Page 9: concerning pointing accuracy: if you observe halos in all segments, you could use combinations of sectors 1+4, 2+5 and 3+6 to better determine the center. Would that improve your accuracy?

This method would in fact increase the accuracy if a 22° halo is visible and pronounced in all 6 segments. The problem is that in the segments 1, 2, 3 the 22° halo is, if visible at all and not obstructed by the horizon, usually much less pronounced than in the upper segments. For a faint halo the peak in the brightness distribution is rather flat causing a larger uncertainty in finding the angular position of the peak. We added the following sentence to the manuscript for clarification:

> "This segment was chosen since it contains the most pronounced halos. For a faint halo the peak in the brightness distribution is rather flat causing a larger uncertainty in finding the angular position of the peak."

Page 10, line 4: A detailed description ... refers to only 9 lines of text. This does not seem very detailed.

We deleted the word "detailed"

Page 10 referring to Sect. A and B should be referring to Appendix A and Appendix B

We replaced Sect. by Appendix.

Caption Fig. 7: says it is the same as Fig. 5: Fig 5 showed R, G, and B. Here there is only one curve. Which one? Or is it something like 1/3(R+B+G) ?

Fig. 7 shows the green channel of the same data as Fig. 5. We changed the caption accordingly.

Page 12, line 2 from bottom: was repeated 100 times. Question: 100 times for the exact same 30%?

The subset items were chosen randomly each time from the training dataset.

I wonder how many halo events were responsible for your halo images. For example if the same cloud gives rise to a 20 minute long halo display, some of your images refer to the same event, i.e. similar ice crystals within the same cloud. It would also be interesting to know the percentage of time a single halo producing cirrus is giving rise to halos. This could also change your statement how many clouds did not produce halos at all. Did you investigate whether there were reproducible differences between the halos of different cirrus clouds from different days (or e.g. if there were some with odd radius halos).

This is a very interesting question. The HaloCam measurements were performed with a time interval of 10 s continuously during day time. We did not cluster the observations to halo "events" since it is not easy to find good criteria for the "same cloud". Even if the camera observes a halo for, say 20 min, different ice crystals are producing it as the cloud is advected by the wind. For example during an approaching warm front a halo may be visible for 20 min in the high cirrus clouds leading the front. Within this time, however, the cirrus cloud base height my already decrease to warmer temperatures where the formation of other ice crystal shapes might dominate. Also the optical thickness of the cirrus might change during the 20 min. So even if there are no gaps in the cloud, should it be considered as the "same cloud"? Maybe such an analysis could be done the other way round by looking at cloud properties, such as height, temperature and optical thickness. I would expect the micro- and macrophysical conditions of a cloud which shows the same halo type for e.g. 20 min to be similar over this time period. This could be an interesting future study using the HaloCam observations together with complementing measurements such as radar, lidar or sunphotometer data.

Page 13: you mention it indirectly but I would point it out more clearly: your test was done by also visually classifying the images. There is of course one problem: the eye may not be good to detect halos on still images. It is much better comparing differences in time sequences. How have you done it? Looking at time lapse sequences or looking at stills? Can you comment?

We agree that the eye is able to detect much fainter halos on time lapse sequences than on still images. HaloForest does not make use of the "optical flow". So this information should also not be used for visually selecting and counter-checking the images. For compiling the training dataset and for counter-checking the classifications the images were randomly selected which is equivalent to looking at still images.

Please comment why you have always used a $2\sigma$ deviation rather than $1\sigma$ or $3\sigma$.

We used a standard deviation of $2\sigma$ since it is often used as an estimate of the uncertainty. The results in Tab. 3 could also be provided with a $1\sigma$ or $3\sigma$ standard deviation. We added the information to the table caption. The $1\sigma$ and $3\sigma$ standard deviation can easily be calculated from the provided $2\sigma$.

"The results are provided with a $2\sigma$ standard deviation."

You mention the false positive or false negative results from your classifier. I expect a discussion. What have you observed when visually counterchecking those images. Could you find some reasons for the results, e.g. some brightly illuminated cloud fractions or shadows or..?

HaloCam images can be mis-classified due to features which look similar to the profile of a 22° halo when averaged azimuthally over the image segment. Scenes which were mis-classified show, for example, a small white cloud on a blue sky located at a scattering angle of 22° or contrails and sometimes small altocumulus clouds which, when averaged azimuthally, exhibit a peak at the 22° halo scattering angle. We added the following sentence:

"Images were incorrectly classified as 22° halo predominantly due to small bright clouds or contrails in a blue sky, or structures in overcast conditions which happen to cause a peak in the averaged brightness distribution at a scattering angle of 22°."

Later on you mention sundogs: I assume for parhelia it would have been better if the segment boundaries would have been rotated such that they would fall within a single segment rather than being split up between 2.

For detecting sundogs I would use a specific mask enclosing only the sundog at the expected position in the image, which depends on the solar zenith angle as in Fig. 1. For an extended algorithm the same image would then be analyzed by two different masks.

Page 13, line 18: no discrimination so far, fine! But please comment whether you think that it will be easily possible to distinguish between the 22° halo and tangent arcs/circumscribed halo.

Upper/lower tangent arcs could be distinguished from the 22° halo for low solar elevations by applying a special mask. For high solar elevations when the tangent arcs merge to the circumscribed halo, a discrimination might be difficult. A detailed description of the detection of other halo types is beyond the scope of this study, but we included some discussion about the possibility to distinguish between upper/lower tangent arcs:

"Upper tangent arcs could be detected by the uppermost image segment (no. 4) and might be mis-classified as 22° halo. For very small SZAs (high solar elevations) the tangent arcs merge to form the circumscribed halo which could be detected in the segments 3 and 5 as well. [...] However, if only a fragment of a halo is visible in the uppermost segment, it is generally difficult to discriminate between an upper tangent arc or circumscribed halo and a 22° halo."

Chapter 4 and your sensitivity study: Nice model but in principle not new. Therefore I miss reference to other relevant work. There are many people who have e.g. applied less sophisticated Monte Carlo methods to halos, also including multiple scattering on halos (I am sure, a proper literature search will show up several papers). There have also been some studies on visibility

of halos with respect to cloud optical thickness from the Atmospheric Optics community (see e.g., Bull. Am. Met. Soc. 89, 471-485 (2008) or AppOpt47/34, H157 (2008)). I do expect a discussion either why you have not mentioned other simulations or you should add some of them and compare your work to their models.

The radiative transfer calculations were only added to demonstrate the qualitative behavior of the 22° halo in a realistic atmosphere. It is beyond the scope of our paper to perform a comparison between different radiative transfer models. Nevertheless, we added a paragraph describing the radiative transfer model and the performed simulations for Gedzelman and Vollmer (2008), Gedzelman (2008) and Kokhanovsky (2008), which was also recommended by referee #2.

"The effect of varying cloud optical thickness on the visibility of halo displays was already investigated by Kokhanovsky (2008); Gedzelman and Vollmer (2008); Gedzelman (2008) using radiative transfer simulations. Kokhanovsky (2008) performed simulations of the brightness contrast of the 22° halo as a function of the cirrus optical thickness using the radiative transfer model SCIATRAN neglecting molecular and aerosol scattering. The results show a linear decrease of the halo contrast with increasing optical thickness. Gedzelman (2008) and Gedzelman and Vollmer (2008) used the model HALOSKY for radiative transfer simulations of halos with varying cloud optical thickness. HALOSKY considers single scattering by air molecules, aerosol particles and cloud particles assuming homogeneous, plane-parallel atmospheric layers. Multiple scattering is calculated only within the cloud by a Monte Carlo subroutine. Gedzelman and Vollmer (2008) show results for radiance simulations of the 22° halo in the principal plane below and above the sun. They found that the radiance at the bottom of the halo reaches a maximum value for smaller COT ($\approx 0.25$) than the radiance at the top of the cloud ($\approx 0.63$)."

Page 15, line 12: skip arc.

Done.

Why did you only use columns, why did you not use any plates in your simulations? Whenever you observe sundogs, you need the plates as well!

We performed the simulations using columns as an example of how the 22° halo is affected by multiple scattering. For randomly oriented particles the effects shown in the manuscript for columns are in principle the same for plates. The main difference would be that a larger fraction of smooth crystals are needed so that a 22° halo is visible. Radiative transfer simulations of sundogs would require oriented particles which are not yet implemented in libRadtran.

Why did you use the spectral albedo for grass. Munich is green I guess, but the institute is probably still in the middle of the town and not only surrounded by parks.

Solar elevations in Munich reach a maximum value of 65°. So the 22° halo region of the observed cirrus clouds are south of Munich, as shown in the following Fig. 3. The surface albedo of this area can well be represented by grass. Since we consider this only a minor aspect, Fig. 3 is only included here.

Page 16: it may be easier to understand and/or helpful to visualize your condition HR=1.0 for a treshhold which can be easily done.

We included markers in Fig. 7 in the manuscript indicating the values of $I(\vartheta_{\text{halo,max}})$ and $I(\vartheta_{\text{halo,max}})$ and added a discussion of the cases HR<1, HR=1, HR>1.

"In analogy, here, the halo ratio (HR) is defined as the brightness $I$ at the scattering angle of the halo maximum $\vartheta_{\text{halo,max}}$ divided by the brightness at the scattering angle of the minimum $\vartheta_{\text{halo,min}}$:

$$\text{HR} = I(\vartheta_{\text{halo,max}})/I(\vartheta_{\text{halo,min}}) \tag{1}$$

As an example, the values for $I(\vartheta_{\text{halo,max}})$ and $I(\vartheta_{\text{halo,min}})$ are displayed in Fig. 2 by the blue triangles pointing up (max) and down (min), respectively. For clearsky conditions and homogeneous cloud cover the brightness

distribution decreases from the sun towards larger scattering angles, as shown in the example in Figs. **??** and 2. If HR $< 1$ the brightness at the scattering angle of the halo maximum ($I(\vartheta_{\mathrm{halo,max}})$) is smaller than for the minimum ($I(\vartheta_{\mathrm{halo,min}})$) which is representative for a monotonically decreasing, featureless curve in this scattering angle region. This is the case for clearsky conditions or homogeneous cloud cover without halo. For HR $= 1$ the brightness at the halo maximum and minimum are the same causing a slight plateau in the brightness distribution. A distinct halo peak occurs for the condition HR $> 1$. Thus, we assume HR $= 1$ as lower threshold for the visibility of a halo."

Page 16, line 14: you mention that in multiple scattering HR decreases. This is plausible, but please also give reference. Or did you only intend to give a qualitative plausibility statement?

The simulations intend only to show qualitatively the effect of multiple scattering on the visibility of the 22° halo. Kokhanovsky (2008) shows similar findings. We included this reference.

"For large COT, multiple scattering reduces the contrast of the halo feature and the HR decreases, similar to the findings of Kokhanovsky (2008). However, as Gedzelman and Vollmer (2008) point out, the halo peak might still be visible up to an optical thickness of $\sim 5$ due to the pronounced maximum in the scattering phase function."

Sect. 5 Conclusions: I miss somewhere – not necessarily here – a discussion about potential reasons for the observed differences in halo frequency observations Sassen/AKM/your work. For example discuss meteorological differences in cloud formation due to different climates etc. You only mentioned a little bit on pages 5,6.

We included Sassen et al. (2003) as reference for the discussion of the meteorological conditions during the presence of the halo displays. However, the correlation between the occurrence of halo displays and certain weather patterns have not been evaluated for the HaloCam observations in Munich and have also not been analyzed for the AKM dataset. The main reason for the differences between our observations and Sassen/AKM is the small temporal range of the observations during the ACCEPT campaing of 6 weeks compared to the long-term observations of Sassen (10 years) and AKM (30 years). We added the following sentences:

"Also differences in the dominating weather patterns forming the cirrus clouds in Salt Lake City and Cabauw could have an impact on halo formation as discussed in Sassen et al. (2003). For the AKM and the HaloCam dataset, information about dominating weather patterns for different halo displays is not available."

I assume – if extending the halo algorithm to parhelia, the color separation may be an additional criterion for distinguishing halo types.

We use information about the color separation already for the detection of 22° halos ($\Delta\vartheta_{\mathrm{halo,min/max}}$). We agree that this might be even more important for detecting parhelia since they have a more distinct separation of colors than the 22° halos.

Outlook: maybe say a few words what may in principle be possible. Can you imagine using a wide angle lens and also detect the colorful circumzenithal arcs which also have a reasonably high observation frequency in Germany?

Using a lens with a wider field of view allows to observe more halo displays of course at the expense of a reduced spatial resolution. See also response to referee #1. We added the following text to the manuscript:

"In principle, HaloCam could also be equipped with a wide-angle lens to observe halo displays in a larger region of the sky, however at the expense of the spatial resolution."

Fig. 11 discussing your decision tree: In the paper you mention and show figures with HR around 1.1 to 1.15, but your criteria in the tree give numbers much lower (1.01 or so). Please comment. You may want to discuss what typical brightness variations under daylight conditions are considered to be easily perceived by the human eye.

Regarding Fig. 11 we state that the tree uses only three of the eight features and is pruned to a depth of three layers. So the criteria used in this case are not the same as used by HaloForest. In general, the halo ratios could well be lower than 1.1 and 1.15 as shown in Fig. 8. When comparing the halo ratios derived from pictures with jpeg compression it should also be kept in mind that the brightness on these images does not change linearly but is compressed by a gamma correction. So the halo ratios could slightly deviate from the brightness contrast in the "raw" signal measured by the camera sensor. We added the following sentence:

> "A typical example of a single decision tree, as used for HaloForest, is shown in Fig. 11. For a better visualization, the tree is grown using only three of the eight features and is pruned to a depth of three layers. The explanation provided here focuses on the structure of tree rather than the exact numbers of the threshold tests which differ from the ones used by HaloForest."

[Figure]

**Figure 1.** HaloCam image as in Fig. 4b. The red and green squares indicate the position of the sundogs as a function of the solar zenith angle (SZA). The SZA ranges between 90° and 35° with 1° resolution. The mask used to search for the 22° halo peak is displayed by the two white circles and covers scattering angles between 21.0° and 23.5°. Sundog positions located within this mask might be mis-classified as 22° halo and are marked as red. These positions correspond with SZAs between 90° and 67°. For smaller SZAs (higher solar elevations) the sundogs are located outside the mask and cannot be mis-classified as 22° halo by the algorithm.

[Figure]

**Figure 2.** As Fig. 5 showing the first minimum (dotted) and the maximum (dashed) of the 22° halo for the green channel. In addition, $\vartheta_{\text{halo,end}}$ is indicated (dash-dot line) which represents the scattering angle of the same brightness as $\vartheta_{\text{halo,min}}$ and confines the halo peak. In this example $\vartheta_{\text{halo,end}}$ is located at about 24.5°. The corresponding brightness $I(\vartheta_{\text{halo,min}})$ and $I(\vartheta_{\text{halo,max}})$ used to calculate the HR are marked with the blue triangles pointing down (min) and up (max). The regression line of the averaged brightness distribution (solid black), which is evaluated between scattering angles of 15° and 30°, has a slope of -2.5 for this example.

[Figure]

**Figure 3.** The blue ellipses represent the 22° halo projected on a cirrus cloud deck at 9 km altitude for different solar positions on 21 March between 9 UTC and 13 UTC in 1 h intervals. The blue pins in the focal point of each ellipse indicates the projected position of the sun. The blue pin a bit north of the center of Munich represents the location of the HaloCam site.

[revised manuscript text omitted]

---

## Author Response (AR2)

**Reply to Associate Editor Decision: Publish subject to minor revisions**

We thank the editor for reviewing the manuscript and comments. We have the impression that most of the editor's comments were already addressed in the revised manuscript and in the replies to the referees' comments uploaded on 30 May 2017 (amt-2017-17-author_response-version2.pdf and amt-2017-17-manuscript-version4.pdf). It might be possible that the editor's comments are based on the original manuscript from March 2017.

In the following we address the editor's comments and included a pdf-version of the manuscript highlighting the new changes according to the editor's comments from 4 June.

Remark: The editor's comments are highlighted in blue. Figure numbers in the authors' reply refer to the figures in the original manuscript. Snippets included in the revised manuscript are highlighted by an additional indent and quotation marks.

Comments to the Author:
Dear Authors
Re manuscript "Ice Crystal Characterization in Cirrus Clouds: ..."
Thankyou for your submission to AMT, which I find to quite an impressive piece of work. There is however a number of concerns that need to be addressed before I can finally accept the manuscript for publication. Could you please address these concerns listed below.
best regards
Murray Hamilton
(Assoc. Ed.)

It appears that referee #3's comments have been ignored - could you please provide a list of how these concerns have been addressed. Many of them are, to my mind, correct and need little editing of the text to address. If you cannot address some of them, you should say why.

For example:
I don't believe that you have adequately addressed the concern of referee #3 about the periods over which data was collected and their respective locations. The abstract says only that the analysed data were collected near Munich, yet section 2.1 refers to data that was analysed and came from Cabauw. Later the paper refers to analysis of data obtained near Munich.

> We addressed the comments of referee #3 already in the Author's Response (amt-2017-17-author_response-version2.pdf), which is structured as follows: the replies to the referees comments are included in numerical order (#1, #2, #3) followed by a manuscript version with highlighted changes. The reply to the comments of referee #3 are on the pdf pages 5–14. We addressed the comment you mention by including the specific time periods for the ACCEPT campaign in the abstract. We clarified this also in section 2.1 (p. 5, lines 5–9, and highlighted changes on p. 15).

Also, referee #1 is correct in that 150.000 should be written as 150,000 - i.e. with a comma. The latter is correct usage in English but this shouldn't be left to the typesetters, as "150.000" means one hundred and fifty, and they won't know what you actually mean.

> We addressed this comment already in amt-2017-17-author_response-version2.pdf (p. 1 and highlighted changes on p. 17).

I also have a few editorial comments ...
You write "As a measure for the colorfulness of the halo ...". "Colorfulness" is not a word in English, though perhaps it should be. Anyway a better way to express this is "As a measure of the separation of color in the halo ..."

> Thank you for pointing this out, we changed the sentence accordingly.

>> "As a measure of the separation of color in the halo, the scattering angle difference between the blue and red channel for the halo minimum ($\Delta\vartheta_{\mathrm{halo,\,min}}$) and maximum ($\Delta\vartheta_{\mathrm{halo,\,max}}$) are calculated [...]"

"Tab. 3 shows the confusion ..." . You should not start a sentence with an abbreviation. This leads me to another issue regarding the tables; the tables need to be connected to the text with references and a short description (in the text). I cannot find this for tables 2 or 4.

We replaced the abbreviation by "Table 3 ..." and changed other references at the beginning of a sentence accordingly. Table 2 is discussed on page 9, lines 2–7 and Tab. 4 on page 14, lines 8–9.

What is "dthteta" in figure 11?

We corrected this to $\Delta$theta_halo_max (see Fig. 12 in the revised manuscript).

Please find below a pdf-version of the manuscript highlighting the changes.

[revised manuscript text omitted]